# Emergent mechanisms for long timescales depend on training curriculum and affect performance in memory tasks

**Sina Khajehabdollahi** [1,2,*], **Roxana Zeraati** [1,3,*], **Emmanouil Giannakakis** [1,3],
**Tim J. Schäfer** [1,3], **Georg Martius** [1,2], **Anna Levina** [1,3]
[1] University of Tübingen, Germany
[2] Max Planck Institute for Intelligent Systems, Tübingen, Germany
[3] Max Planck Institute for Biological Cybernetics, Tübingen, Germany
[*] These authors contributed equally to this work.
{firstname.lastname}@uni-tuebingen.de

## Abstract

Recurrent neural networks (RNNs) in the brain and *in silico* excel at solving tasks with intricate temporal dependencies. Long timescales required for solving such tasks can arise from properties of individual neurons (single-neuron timescale, $\tau$, e.g., membrane time constant in biological neurons) or recurrent interactions among them (network-mediated timescale, $\tau_{\text{net}}$). However, the contribution of each mechanism for optimally solving memory-dependent tasks remains poorly understood. Here, we train RNNs to solve $N$-parity and $N$-delayed match-to-sample tasks with increasing memory requirements controlled by $N$, by simultaneously optimizing recurrent weights and $\tau$s. We find that RNNs develop longer timescales with increasing $N$, but depending on the learning objective, they use different mechanisms. Two distinct curricula define learning objectives: sequential learning of a single-$N$ (single-head) or simultaneous learning of multiple $N$s (multi-head). Single-head networks increase their $\tau$ with $N$ and can solve large-$N$ tasks, but suffer from catastrophic forgetting. However, multi-head networks, which are explicitly required to hold multiple concurrent memories, keep $\tau$ constant and develop longer timescales through recurrent connectivity. We show that the multi-head curriculum increases training speed and stability to perturbations, and allows generalization to tasks beyond the training set. This curriculum also significantly improves training GRUs and LSTMs for large-$N$ tasks. Our results suggest that adapting timescales to task requirements via recurrent interactions allows learning more complex objectives and improves the RNN's performance.

## 1 Introduction

The interaction of living organisms with their environment requires the concurrent processing of signals over a wide range of timescales, from short timescales of coding sensory stimuli (Bathellier et al., 2008; Panzeri et al., 2010; Safavi et al., 2023) to longer timescales of cognitive processes like working memory (Jonides et al., 2008). The diverse timescales of these tasks are reflected in the dynamics of the neural populations performing the corresponding computations in the brain (Murray et al., 2014; Cavanagh et al., 2020; Gao et al., 2020; Zeraati et al., 2022). At the same time, artificial neural networks performing memory-demanding tasks (speech (Graves et al., 2013), handwriting (Graves, 2013), sketch (Ha & Eck, 2018), language (Bowman et al., 2015), time series prediction (Chung et al., 2014; Torres et al., 2021), music composition (Boulanger-Lewandowski et al., 2012)) need to process the temporal dependency of sequential data over variable timescales. Recurrent neural networks (RNNs) (Elman, 1990; Hochreiter & Schmidhuber, 1997; Lipton et al., 2015; Yu et al., 2019) have been introduced as a tool that can learn such temporal dependencies using back-propagation through time.

In biological networks, diverse neural timescales emerge via a variety of interacting mechanisms. Timescales of individual neurons in the absence of recurrent interactions are determined by cellu-

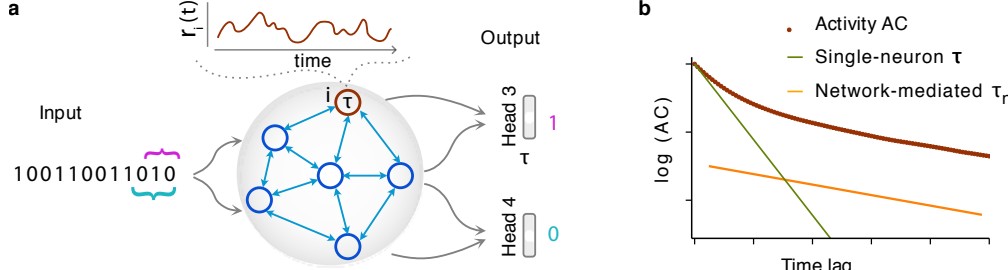

Figure 1: Schematics of network structure and timescales. **a.** An outline of the network. A binary sequence is given as input to a leaky RNN, with each neuron's intrinsic timescale being a trainable parameter $\tau$. The illustration shows the $N$-parity task with readout heads for $N = 3$ and $N = 4$. **b.** An illustration of the manifestation of different timescales (single-neuron and network-mediated) on the autocorrelation (AC) of a network neuron (see also Fig. S2).

lar and synaptic processes (e.g., membrane time constant) that vary across brain areas and neuron types (Gjorgjieva et al., 2016; Duarte et al., 2017). However, recurrent interactions also shape neural dynamics introducing network-mediated timescales. The strength (Ostojic, 2014; Chaudhuri et al., 2015; van Meegen & van Albada, 2021) and topology (Litwin-Kumar & Doiron, 2012; Chaudhuri et al., 2014; Zeraati et al., 2023; Shi et al., 2023) of recurrent connections give rise to network-mediated timescales that can be much longer than single-neuron timescales.

Heterogeneous and tunable single-neuron timescales have been proposed as a mechanism to adapt the timescale of RNN dynamics to task requirements and improve their performance (Perez-Nieves et al., 2021; Tallec & Ollivier, 2018; Quax et al., 2020; Yin et al., 2020; Fang et al., 2021; Smith et al., 2023b; Jain et al., 2020). In these studies, the time constants of individual neurons are trained together with network connectivity. For tasks with long temporal dependencies, the distribution of trained timescales becomes heterogeneous according to the task's memory requirements (Perez-Nieves et al., 2021). Explicit training of single-neuron timescales improves network performance in benchmark RNN tasks in rate (Tallec & Ollivier, 2018; Quax et al., 2020) and spiking (Yin et al., 2020; Fang et al., 2021; Perez-Nieves et al., 2021) networks and leads to greater robustness (Perez-Nieves et al., 2021) and adaptability to novel stimuli (Smith et al., 2023b). While these studies propose the adaptability of single-neuron timescales as a mechanism for solving time-dependent tasks, the exact contribution of single-neuron and network-mediated timescales in solving tasks is unknown.

Here, we study how single-neuron and network-mediated timescales shape the dynamics and performance of RNNs trained on long-memory tasks. We show that the contribution of each mechanism in solving such tasks largely depends on the learning objective defined by the curriculum. Challenging common beliefs in the field, we identify settings where trainable single-neuron timescales offer no advantage in solving temporal tasks. Instead, adapting RNNs' timescales using network-mediated mechanisms improves training speed, stability and generalizability.

## 2 MODEL

We approximate the effect of the membrane timescale of biological neurons by equipping each RNN-neuron with a trainable leak parameter $\tau$, defining the single-neuron timescale (Fig. 1a). The activity of each neuron evolves over discrete time steps $t$ governed by:

$$r_i(t) = \left[\left(1 - \frac{\Delta t}{\tau_i}\right) \cdot r_i(t - \Delta t) + \frac{\Delta t}{\tau_i} \cdot \left(\sum_{j \neq i} W_{ij}^R \cdot r_j(t - \Delta t) + W_i^I \cdot S(t) + b^R + b^I\right)\right]_\alpha,$$

(1)

where $[\cdot]_\alpha$ is the leaky ReLU function with negative slope $\alpha$, given by:

$$[x]_\alpha = \begin{cases} x, & x \geq 0 \\ \alpha \cdot x, & x < 0. \end{cases}$$

(2)

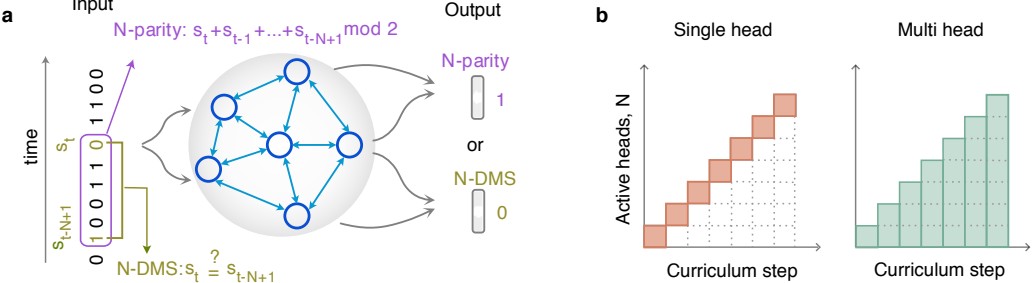

Figure 2: Schematic description of the tasks and curricula **a.** An outline of the network and tasks. In both tasks, the network receives a binary input sequence, one bit at each time step. **b.** In the single-head curriculum, only one read-out head is trained at each curriculum step, while in the multi-head curriculum, a new read-out head is added at each step without removing the older heads.

For all networks, we use $\alpha = 0.1$. We obtain similar results using a different type or location of nonlinearity (Appendix A). $W^R, b^R$, and $W^I, b^I$ are the recurrent and input weights and biases, respectively, $S$ is the binary input given to the network at each time step, and $\tau_i \geq 1$ is the trainable timescale of the neuron. Unless otherwise stated, the time step is $\Delta t = 1$ (other $\Delta t$ discussed in Appendix B). Each RNN has 500 neurons. When $\tau = 1$, a neuron becomes memory-less (in isolation) as its current state does not directly depend on its past activity, i.e., memory can only be stored at the network level via interactions. In contrast, for $\tau > 1$, the neuron's activity depends on its past activity, and the dependency increases with $\tau$. In the limit of $\tau \to \infty$, the neuron's activity is constant, and the input has no effect.

The dynamics of each neuron can be characterized by two distinct timescales: (i) single neuron timescale $\tau$, (ii) network mediated timescale $\tau_{\text{net}}$. $\tau$ gives the intrinsic timescale of a neuron in the absence of any network interaction, while $\tau_{\text{net}}$ is shaped by the combination of $\tau$ and the learned connectivity and represents the effective timescale of the neuron's activity within the network. $\tau_{\text{net}}$ is generally a function of $\tau$ and recurrent weights: $\tau_{\text{net}} = f(\tau, W^R)$(Ostojic, 2014; Chaudhuri et al., 2014; Shi et al., 2023) and $\tau_{\text{net}} \geq \tau$ (Shi et al., 2023). For networks with linear dynamics, $\tau_{\text{net}}$ can be directly estimated from the eigenvalues of the connectivity matrix normalized by $\tau$ (Chaudhuri et al., 2014). For nearest-neighbor connectivity or mean-field dynamics, it is also possible to derive $\tau_{\text{net}}$ analytically for nonlinear networks. However, a general analytical solution does not exist. Instead, $\tau_{\text{net}}$ can be effectively estimated from the decay rate of the autocorrelation (AC) function. The AC is defined as the correlation coefficient between the time series and its copy, shifted by time $t'$, called the time lag. For the neuron's activity, it can be computed as

$$\text{AC}_i(t') = \frac{1}{\hat{\sigma}_i^2(T - t')} \sum_{t=0}^{T-t'} \left(r_i(t) - \hat{\mu}_i\right)\left(r_i(t - t') - \hat{\mu}_i\right), \tag{3}$$

where $\hat{\mu}_i$ and $\hat{\sigma}_i^2$ are the sample mean and variance of $r_i(t)$. To estimate $\tau_{\text{net}}$, we drive the network by uncorrelated binary inputs sampled from a Bernoulli distribution.

The AC of a neuron's activity, defined by Eq. 1, can be approximated by two distinct timescales which appear as two slopes in logarithmic-linear coordinates (Fig. 1b) (Shi et al., 2023). The steep initial slope indicates $\tau$, and the shallower slope indicates $\tau_{\text{net}}$. In the same way, we characterize the timescale of collective network dynamics by computing population activity (summed activity of all neurons within a network) timescale $\tau_{\text{pop}}$, which reflects the timescale of network dynamics as a whole. To avoid AC bias in our estimates (Zeraati et al., 2022), we use long simulations ($T = 10^5$ time steps). We simulate each network for 10 trials (i.e. 10 distinct realizations of inputs) and compute the average AC of each neuron across trials. To estimate $\tau_{\text{net}}$, we fit the average AC with a single- ($\tau_{\text{net}} = \tau$) and with a double-exponential ($\tau_{\text{net}} > \tau$) decay function using the nonlinear least-squares method. Then, we use the Akaike Information Criterion (AIC) (Akaike, 1974) to select the best-fitting model. For most neurons (above 95%), Bayesian information criterion (BIC) selects the same model (Fig. S1) and previous work (Pasula, 2023) indicates that AIC provides similar results as the sum of three information criteria (AIC, BIC and Hannan-Quinn information criteria).

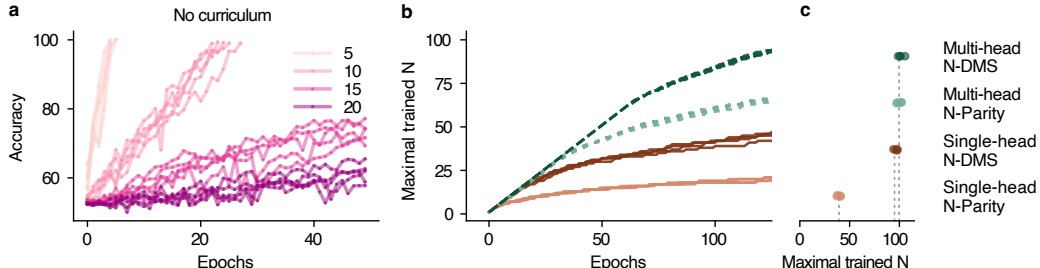

Figure 3: Training performance depends on the curriculum. **a.** Accuracy of training the networks ($N$-parity task) without a curriculum increases slowly, especially when $N > 10$. For each $N$, 5 models are independently trained for 50 epochs or until reaching $> 98\%$ accuracy. **b.** Multi-head (dashed) trained networks are solving larger $N$s than single-head (solid) within the same training time (colors in c). **c.** The maximum trained $N$ for each task/curriculum at the end of training (1000 epochs or solving $N = 101$, whichever comes first). Gray lines - mean value across 4 networks.

When the double-exponential model is selected, the slowest of two timescales indicates $\tau_{\text{net}}$. For most fits, we obtain a large coefficient of determination, confirming a good quality of fit (Fig. S2).

## 3 SETUP

### 3.1 TASKS

In both tasks (Fig. 2a), a binary sequence $S$ is given as the input, one bit at each time step. We train the networks on sequences with lengths uniformly chosen from the interval $L \in \{N + 2, 4N\}$.
**$N$-delayed match-to-sample ($N$-DMS):** The network outputs 1 or 0 to indicate whether the digit presented at current time $t$ matches the digit presented at time $t - N + 1$. To update the output at every time step, the network needs to store the values and order of the last $N$ digits in memory.
**$N$-parity:** The network outputs the binary sum (XOR) of the last $N$ digits. $N$-parity has a similar working memory component as $N$-DMS, but requires additional computations (binary sum).

### 3.2 TRAINING

We train single-neuron timescales $\tau = \{\tau_1, \ldots, \tau_{500}\}$, $W^R, b^R, W^I, b^I$, and a linear readout layer via back-propagation through time using a stochastic gradient descent optimizer with Nesterov momentum and a cross-entropy loss. Each RNN is trained on a single Nvidia GeForce 2080ti for 1000 epochs, 3 days, or until the $N = 101$ task is solved, whichever comes first. RNNs are trained without any regularization. Including L2 regularization achieves comparable performance.

**Single-head:** Starting with $N = 2$, we train the network to reach an accuracy of $98\%$. We then use the trained network parameters to initialize the next network that we train for $N + 1$ (Fig. 2b).
**Multi-head:** As with the single-head networks, we begin with a network solving a task for $N = 2$, but once a threshold accuracy of $98\%$ is reached, a new readout head is added for solving the same task for $N + 1$, preserving the original readout heads. At each curriculum step, all readout heads are trained simultaneously (the loss is the sum of all readout heads' losses) so that the network does not forget how to solve the task for smaller $N$s (Fig. 2b).

## 4 RESULTS

### 4.1 PERFORMANCE UNDER DIFFERENT CURRICULA

**Necessity of curriculum**: Our objective is to learn the largest possible $N$ in each task. First, we test whether a good performance can be achieved for high $N$ in either task without any curricula. We find that for both tasks (see Appendix E for $N$-DMS results), networks struggle to reach high accuracy for $N > 10$ (Fig. 3a, Fig. S13). However, using either curriculum significantly boosts the network's capacity to learn tasks with larger $N$ (Fig. 3b).

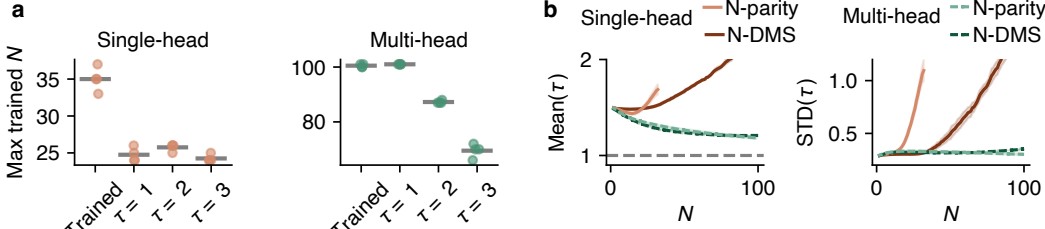

Figure 4: Importance of single-neuron timescales for different curricula. **a.** The maximum $N$ solved in the $N$-bit parity task after 1000 epochs (reaching an accuracy of $98\%$). X-label indicates training constraints: $\tau = 1, 2$ or $3$ - fixed $\tau$ with only weights being trained, "Trained" allows training of $\tau$. In the single-head curriculum, models rely on training $\tau$, whereas in the multi-head curriculum, $\tau$ fixed $\tau = 1$ is as good as training $\tau$. Horizontal bars - mean. **b.** The mean and standard deviation (STD) of the trained $\tau$s increase with $N$ in single-head networks. In contrast, in multi-head networks, the mean $\tau$ decreases towards 1, and the STD remains largely constant. The mean and STD are computed across neurons within each network (up to the maximum $N$ shared between all trained networks). Shading - variability across 4 trained networks.

**Comparison of curricula**: Task performance differs significantly between curricula. The single-head networks can reliably reach $N \approx 35$ for the $N$-parity and $N \approx 90$ for the $N$-DMS task. The difference in performance between the two tasks is expected because the $N$-DMS task is much easier than the $N$-parity task. However, the multi-head networks can reliably reach $N \geq 100$ for both tasks and require fewer training epochs to reach $98\%$ accuracy for each $N$ (Fig. 3b, c). Networks that are trained using an intermediate curriculum between the two extremes of single- and multi-head (i.e., solving simultaneously $H < N$ tasks for $N, N - 1, \ldots, N - H + 1$ with gradually increasing $N$s) exhibit an intermediate performance (Appendix F). Overall, the multi-head networks outperform the single-head ones in terms of performance (maximum $N$ reached) and the required training time. Moreover, the multi-head curriculum significantly improves the training of other recurrent architectures such as GRU and LSTM to perform large-$N$ tasks (Appendix G), suggesting that the multi-objective curriculum can generally improve learning long-memory tasks.

The superior performance of the multi-head networks may be counter-intuitive since they solve the task for every $N \leq m$ at the $m$-th step of the curriculum, whereas the single-head networks only solve it for exactly $N = m$. However, we find that the single-head networks suffer from catastrophic forgetting: a network trained for larger $N$ cannot perform the task for smaller $N$s it was trained for, even after retraining the readout weights (Appendix H). These results suggest that explicit prevention of catastrophic forgetting by learning auxiliary tasks (i.e. tasks with $N$ smaller than objective, $N < m$) facilitates learning large $N$s. Interestingly, training directly on a multi-$N$ task without using an explicit curriculum results in the emergence of the multi-head curriculum: networks learn to first solve small-$N$ tasks and then large-$N$ tasks (Appendix I), supporting the use of the multi-head step-wise strategy.

**Necessity of training $\tau$**: We examine the impact of training single-neuron timescales on training performance. We compare the training performance of networks with a fixed $\tau \in \{1, 2, 3\}$ shared across all neurons versus networks with trainable timescales. In the single-head curriculum, the training performance with fixed $\tau$ is significantly worse than when we train $\tau$ (Fig. 4a). On the other hand, In multi-head networks, training performance is the same for fixed $\tau = 1$ and trainable $\tau$ cases, but steeply declines for fixed $\tau \geq 2$ (Fig. 4b). See (Fig. S10) for $N$-DMS results. These results indicate that single-head networks rely on $\tau$ for solving the task, whereas multi-head networks only use $\tau$ to track the timescale of updating the input and rely on other mechanisms to hold the memory.

## 4.2 MECHANISMS UNDERLYING LONG TIMESCALES

To uncover the mechanisms underlying the difference between the two curricula, we study how $\tau$, $\tau_{\text{net}}$, $\tau_{\text{pop}}$ and recurrent weights change with increasing task difficulty $N$. One can expect that as the timescale of the task (mediated by $N$) increases, neurons would develop longer timescales to integrate the relevant information. Such long timescales can arise either by directly modulating $\tau$ for each neuron or through recurrent interactions between neurons reflected in $\tau_{\text{net}}$ and $\tau_{\text{pop}}$.

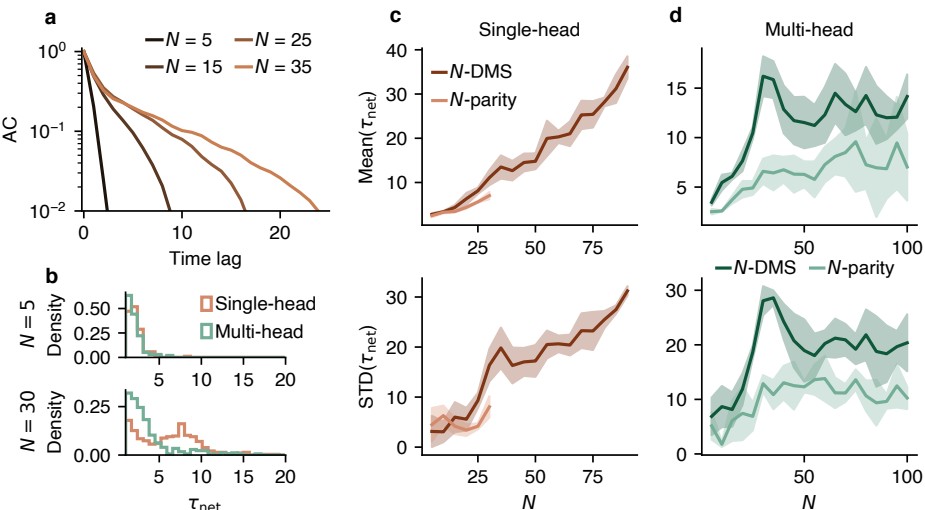

Figure 5: The emergence of network-mediated timescales depends on the curriculum. **a.** Example average ACs of all the neurons within a single-head network, $N$-parity task. The ACs of individual neurons' activity decay slower with increasing $N$. **b.** Distributions of the network-mediated timescales $\tau_{\text{net}}$ for single and multi-head networks solving $N$-parity task for $N = 5$ and $N = 30$. The distribution becomes broader for higher $N$. **c, d.** The mean and STD of the network-mediated timescale $\tau_{\text{net}}$ increase with $N$ in both tasks. The mean and STD are computed across neurons within each network. Shades - variability across 4 trained networks.

**Dependence of $\tau$ on $N$:** The two curricula adjust their $\tau$ to $N$ in distinct ways: single-head networks increase their $\tau$ with $N$, but multi-head networks prefer $\tau \to 1$ (Fig. 4b). For the single-head curriculum, the mean and variance of $\tau$ increase with $N$, suggesting that not only $\tau$s become longer as the memory requirement grows, but they also become more heterogeneous. We obtain similar results for networks trained without curriculum (Fig. S3). On the contrary, in multi-head networks, the average $\tau$ decreases with $N$, approaching $\tau = 1$. The trend of $\tau \to 1$ is consistent with the fact that multi-head networks with fixed $\tau = 1$ performed as well as networks with trained $\tau$.

**Dependence of $\tau_{\text{net}}$ and $\tau_{\text{pop}}$ on $N$:** Network-mediated timescales $\tau_{\text{net}}$ and $\tau_{\text{pop}}$ generally increase with $N$ in both curricula (Fig. 5, Fig. S4). $\tau_{\text{net}}$ reflects the contribution of $\tau$ and recurrent weights in dynamics of individual neurons. In single-head networks, the mean and variance of $\tau_{\text{net}}$ follow a similar trend as $\tau$ (Fig. 5c), suggesting that changes in $\tau_{\text{net}}$ can arise from changes in $\tau$. The mean and variance of $\tau_{\text{net}}$ in multi-head networks increase with $N$ up to some intermediate $N$, but the pace of increase reduces gradually and saturates for very large $N$s (Fig. 5d, top). Given the small $\tau$ in multi-head networks, long $\tau_{\text{net}}$ can only arise from recurrent interactions between neurons. $\tau_{\text{pop}}$ is the timescale of collective network dynamics (sum of all neurons' activations) and arises from interactions between neurons within the whole network. $\tau_{\text{pop}}$ exhibits a clear increase with $N$ for both tasks and curricula with comparable values (Fig. S4). These results indicate that in both curricula, collective network dynamics become slower with increasing $N$, but due to differences in $\tau$, the two curricula employ distinct mechanisms to achieve this.

**Dependence of connectivity on $N$:** Multi-head networks have, on average, almost the same total incoming positive and negative weights (with a slight tendency towards larger total negative weights as $N$ increases), leading to relatively balanced dynamics (Fig. 6a). On the other hand, single-head networks have a stronger bias towards more inhibition (negative weights) as $N$ increases. The strong negative weights in single-head networks are required to create stable dynamics in the presence of long single-neuron timescales (Appendix J).

**Dimensionality of dynamics:** The dimensionality of population activity (measured as the number of principal components that explain $90\%$ of the variance) increases almost linearly with $N$ in the $N$-parity task but sub-linearly in the $N$-DMS task, using both curricula, reflecting distinct computational requirements for each task (Fig. 6b, S20, Appendix K). Since computations should be

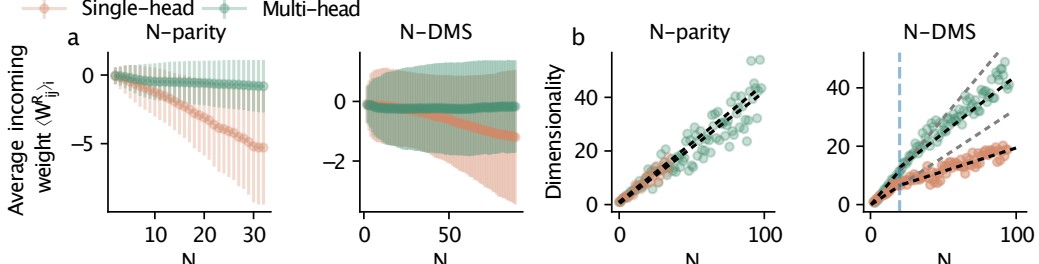

Figure 6: Learned recurrent connectivity and dimensionality of population activity. **a.** The average incoming weight to a neuron remains close to zero for multi-head networks but becomes strongly negative as $N$ increases in single-head networks (example RNN). Error bars - $\pm$ STD. **b.** The dimensionality of the activity increases linearly with $N$ in $N$-parity and sub-linearly in $N$-DMS task. Dashed lines - linear fit computed for all $N$s ($N$-parity task) and independently for $N \in [0, 20]$ (gray line – extension for visual guidance) and $N \in [20, 100]$ ($N$-DMS task). Blue line - $N = 20$.

performed at every time step, the increase in dimensionality may be required to map different input patterns (that grow with $N$) to the same outputs.

Our findings suggest that both curricula give rise to networks with slower and higher dimensional collective dynamics with increasing $N$, but via distinct mechanisms. In single-head networks, single-neuron properties $\tau$ play an important role in creating slow dynamics, which are then stabilized by stronger inhibition in the network. However, in multi-head networks, the slow dynamics should arise from recurrent network interactions. The significant difference in performance of the two curricula suggests that the second mechanism is more effective in solving the task.

### 4.3 IMPACT OF DIFFERENT CURRICULA ON NETWORKS ROBUSTNESS

To compare the robustness and retraining capability between the two curricula, we investigate changes in network accuracy resulting from ablations, perturbations, and retraining networks on unseen $N$. We measure the effects on network performance using a relative accuracy metric with respect to the originally trained network, defined as $\mathrm{acc}_{\mathrm{rel}} := (\mathrm{acc} - 0.5)/(\mathrm{acc}_{\mathrm{base}} - 0.5)$, where $\mathrm{acc}_{\mathrm{base}}$ represents the accuracy of the network before any perturbations or ablations, $\mathrm{acc}$ is the measured accuracy after the intervention, and $0.5$ is a chance level used for the normalization. If $\mathrm{acc}_{\mathrm{rel}} = 1$, the intervention did not change the accuracy; when $\mathrm{acc}_{\mathrm{rel}} \approx 0$, the intervention reduced the accuracy to chance level. All accuracies are evaluated on the maximal trained $N$.

**Ablation:** To examine the relative impact of neurons with different trained $\tau$ on network performance, we ablate individual neurons based on their $\tau$ and measure the performance without retraining (Appendix L). Specifically, we compare the effect of ablating the 20 longest ($4\%$ of the network) and 20 shortest timescale neurons from the network (Fig. 7a,b). For small $N$ and both curricula, ablating individual neurons has only minimal effect (less than $1\%$) on accuracy. However, for larger $N$, we observe a considerable difference in the importance of neurons. Single-head networks rely strongly on long-timescale neurons (Fig. 7a), such that ablating them reduces the performance much more than for short-timescale neurons. In contrast, multi-head networks exhibit greater robustness against ablation, and their accuracy is more affected when short-timescale neurons (i.e., neurons with $\tau = 1$) are ablated (Fig. 7b). Note that in single-head networks, the average of the 20 longest single-neuron timescales is 2.7 times longer than in multi-head networks.

**Perturbation:** We perturb $W^R$ and $\tau$ with strength $\varepsilon$ as (Wu et al., 2020)

$$\widetilde{W}^R = W^R + \varepsilon \frac{\xi_W}{||\xi_W||} ||W^R||, \quad \tilde{\tau} = \tau + \varepsilon \left| \frac{\xi_\tau}{||\xi_\tau||} \right| ||\tau||, \quad \xi_W \sim \mathcal{N}(0, \mathbb{I}^{n \times n}), \quad \xi_\tau \sim \mathcal{N}(0, \mathbb{I}^n).$$
(4)

$|| \cdot ||$ represents Frobenius norm and $| \cdot |$ the absolute value. $\tau$ is perturbed positively to avoid $\tau < 1$. Multi-head networks are more robust to perturbations (Fig. 7c,d). The robustness to changes in $W^R$ is noteworthy since these networks rely on connectivity to mediate long timescales.

**Retraining:** We evaluate the performance of networks that solve $N = 16$, when retrained without curriculum (without training on intermediate $N$s) for an arbitrary higher $N$, after 20 epochs. Multi-

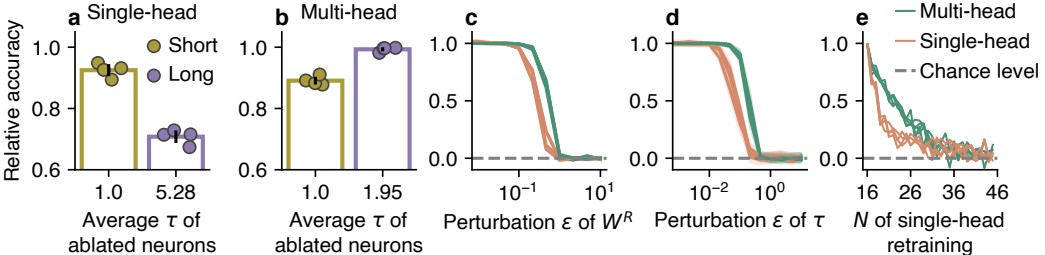

Figure 7: Multi-head networks are more robust to ablation and perturbation, and better retrainable. **(a, b)** Ablating long-timescale neurons largely decreases the performance of single-head networks (a), while multi-head networks (b) are more affected by the ablation of short-timescale neurons ($N = 30$). **(c, d)** Multi-head networks are more robust against perturbations of recurrent connectivity $W^R$ and $\tau$ than single-head networks (note the log-scale x-axis, $N = 30$). **(e)** Multi-head networks retrain faster: They achieve higher relative accuracy when retrained for 20 epochs without a curriculum for a higher $N$. Networks trained to solve $N = 16$ are retrained for 20 epochs to solve higher $N$ without a curriculum to compare re-trainability. Bars - mean; dots and lines - 4 networks for each curriculum; error bars and shades - $\pm$ STD.

head networks show a superior retraining ability compared to single-head networks for at least 10 $N$ beyond $N = 16$ (Fig. 7e). This finding suggests that multi-head networks are better at learning the underlying task and adjust faster to a new, larger $N$ even when skipping intermediate $N$s.

## 5 RELATION TO NEUROSCIENCE

**Continuous-time setting:** So far, we described the tasks and network dynamics in discrete time with time-step $\Delta t = 1$. However, to make the connection to more realistic settings, such as neuroscience, we need to describe the dynamics in continuous time. For this purpose, we consider that each input digit is presented to the network for a certain time $T$. In neuroscience experiments (e.g., DMS task), $T$ is often set to 250-500 ms (Meyer et al., 2011; Qi et al., 2011; Kim & Sejnowski, 2021).

First, we show that for a fixed $T$, the discretization time step $\Delta t$ does not affect the networks' performance and dynamics. We train the networks with each curriculum using a variety of $\Delta t$ (Appendix B). In these networks, the performance on the test data is comparable for different $\Delta t$, even for values much smaller than what was included during the training, mimicking continuous-time dynamics (Fig. S5 c-d). Moreover, similar to our previous results, we find that single-head networks increase their $\tau$ with $N$, while multi-head networks try to reach $\tau \to T$ (Fig. S5 a,b).

Next, we test how changes in input presentation time $T$ affect the dynamics. We set $T = k\Delta t$, where $\Delta t = 1$. We find that for all $k$, single-head networks increase $\tau$ with $N$, whereas multi-head networks' $\tau$ tends to $k$, thus matching the timescales of the input changes (Appendix C).

**Timescales and learning in the biological neural networks:** Our findings suggest that networks that are required to solve tasks with larger memory requirements should develop longer timescales. This largely agrees with findings in the brain: higher cortical areas that are involved in cognitive processes with larger memory requirements (e.g., working memory, evidence accumulation) have longer timescales than sensory areas (Murray et al., 2014). Moreover, we show that developing longer network timescales via changes in network connectivity is a superior solution (in terms of performance and stability) than using longer single-neuron timescales. This is consistent with findings that neural timescales in primate visual cortex adapt to task demands via recurrent network interactions rather than biophysical time constants (Zeraati et al., 2023). Moreover, this result aligns with the learning strategy of biological neural networks, which primarily relies on changes in synaptic strengths rather than modifying the biophysical time constants of individual neurons. While such changes do happen in biology (via protein turnover (Sun & Schuman, 2022), calcium currents (Tiganj et al., 2015), and other mechanisms ), synaptic strength modification is overwhelmingly the mechanism by which biological networks learn.

## 6 RELATED WORK

Previous works independently investigated the role of neuronal and network-mediated timescales in solving memory tasks and proposed inconsistent solutions. Studies focusing on neuronal aspect suggested heterogeneous and adaptable neuronal properties (e.g., membrane time constant) as an optimal mechanism (Perez-Nieves et al., 2021; Mahto et al., 2021; Smith et al., 2023a; Quax et al., 2020). At the same time, other studies presented that network-mediated mechanisms like balanced dynamics (Lim & Goldman, 2013), strong inhibition (Kim & Sejnowski, 2021) or homeostatic plasticity (Cramer et al., 2020; 2023) can create timescales required for memory tasks. For a single neuron modeled with multiple memory units, long timescales were shown to be instrumental in solving memory tasks (Spieler et al., 2023). Here, we explicitly compare these mechanisms and show that while both can be useful for learning long-memory tasks, applying network-mediated mechanisms leads to faster training and more robust solutions.

We find that the difference between mechanisms is revealed mainly in the context of distinct learning objectives defined by curricula. This is an important distinction with previous work, since the role of timescales has been often studied when RNNs solve a single task (e.g., single-head DMS), without considering learning dynamics or the potential for catastrophic forgetting. We relate the mechanisms of task-dependent timescale with the learning dynamics of RNNs across curricula. The use of curricula in our study is inspired by previous work suggesting curriculum learning as a fitness landscape-smoothing mechanism that can enable the gradual learning of highly complex tasks (Elman, 1993; Bengio et al., 2009; Krueger & Dayan, 2009) and be used to uncover distinct learning mechanisms (Kepple et al., 2022; Dekker et al., 2022). Here, we extend these findings by demonstrating how different curricula can push networks towards adopting different strategies to develop slow collective dynamics required for solving long-memory tasks.

## 7 DISCUSSION

We find that to solve long-memory tasks, RNNs develop high-dimensional activity with slow timescales via two distinct combinations of connectivity and single-neuron timescales. While single-head networks crucially rely on the long single-neuron timescales to perform the task, multi-head networks prefer a constant single-neuron timescale and solve the task relying only on the long timescales emerging from recurrent interactions. We show that developing long timescales via recurrent interactions instead of single-neuron properties is optimal for learning memory tasks and leads to more stable and robust solutions, which can be a beneficial strategy for brain computations.

Our findings suggest that training networks on *sets* of related memory tasks instead of a single task improves performance and robustness. By progressively shaping the loss function with a curriculum to include performance evaluations on sub-tasks that are known to correlate with the desired task, we can smooth the loss landscape of our network to allow training for difficult tasks that were previously unsolvable. In this way, choosing an appropriate curriculum can act as a powerful regularization.

**Limitations:** Our study considers two relatively simple tasks with explicitly controllable memory requirements. In follow-up studies, it would be important to test our observations in more sophisticated tasks and investigate whether our results apply to other architectures and optimizers. Additionally, our approach is suitable only for a set of tasks with controllably increasing memory requirements, where the different versions of the same task can be simultaneously performed on the same data (multi-head training). This is a relatively strong constraint, and future research expanding our findings could focus on generalizing the multi-head curriculum for training more realistic tasks. Time series reconstruction is a potential task that can be used to uncover generative dynamical systems from data (Durstewitz et al., 2023). We proposed a potential experiment in Appendix D.

Our current model is a crude approximation of biological neural networks, and more plausible architectures (spiking models, distinct neuron types) could be studied. Finally, biological neural networks can produce long timescales via various other mechanisms we did not consider here (short-term plasticity (Hu et al., 2021), adaptation (Salaj et al., 2021; Beiran & Ostojic, 2019), synaptic delays, etc.). A follow-up study could investigate whether our findings extend to more plausible networks incorporating such additional mechanisms.

## ACKNOWLEDGMENTS

This work was supported by a Sofja Kovalevskaja Award from the Alexander von Humboldt Foundation, endowed by the Federal Ministry of Education and Research (SK, RZ, EG, AL), the Deutsche Forschungsgemeinschaft (DFG, German Research Foundation) under Germany's Excellence Strategy - EXC number 2064/1 - Project number 390727645 (RZ, EG), and Else Kröner Medical Scientist Kolleg "ClinbrAIn: Artificial Intelligence for Clinical Brain Research" (TJS). We acknowledge the support from the BMBF through the Tübingen AI Center (FKZ: 01IS18039B), International Max Planck Research School for the Mechanisms of Mental Function and Dysfunction (IMPRS-MMFD), and International Max Planck Research School for Intelligent Systems (IMPRS-IS). We thank Victor Buendía for valuable discussions.

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

APPENDIX

## A  DIFFERENT TYPES AND LOCATIONS OF NONLINEARITY

In order to verify that our results are robust with respect to the type of nonlinearity used in the network, we train RNNs using two of the most commonly used nonlinearities: ReLU and Tanh. We find that in both cases, the training performance is similar to leaky ReLU, and the development of single-neuron and network-mediated timescales follow the same trajectory as N increases (Fig. S7).

In some implementations of leaky-RNN, the neural self-interaction is linear and located outside of the nonlinearity (cf. equ.1)

$$r_i(t) = \left(1 - \frac{\Delta t}{\tau_i}\right) \cdot r_i(t - \Delta t) + \left[\frac{\Delta t}{\tau_i} \cdot \left(\sum_{i \neq j} W_{ij}^R \cdot r_j(t - \Delta t) + W_i^I \cdot S(t) + b^R + b^I\right)\right]_\alpha . \quad (5)$$

with the explicit time discretization $\Delta t$. The input is presented for time duration $T = k\Delta t$ with input-update time steps $k$. In the main text, we chose $k = 1$ and $\Delta t = 1$. We discuss $k > 1$ in Appendix C and $\Delta t < 1$ in Appendix B.

We verify that training RNNs with this implementation gives similar training dynamics and trajectories of $\tau$ and $\tau_{\text{net}}$ with increasing $N$ (Fig. S8), for both curricula. Furthermore, we find that, for large $N$, ablating neurons with long $\tau$ in single-head networks and neurons with short $\tau$ in multi-head networks reduces the performance significantly, compatible with the findings in the main text (Fig. S9, cf. Fig. 7). We also verify that the performance of the model depends on the initialization of $\tau$ and its trainability in the same way regardless of the location of the nonlinearity (Fig. S11, cf. Fig. 4).

## B  CHANGING TIME DISCRETIZATION

In computational neuroscience, the single neuron dynamics are typically captured by the differential equations that need to be discretized for running numerical simulations and training networks. However, the discretization can be important for stability and internal representation of the model and the task. In the main text, we used Eq. 5 with $\Delta t = T = 1$. For simplicity of notation, we take in the rest of this section $T = 1$. We train networks with different values of $\Delta t$ (a different $\Delta t$ for each training batch), so they can perform the same task independent time discretization. We take $\Delta t = 1/n$ with $n \in \mathbb{N}$ and train the network while keeping the duration of each stimulus presentation in units of time constant (which means that with larger $n$, it would be presented for more time steps). The flexible framework for time discretization allows us to train with multiple $\Delta t$ simultaneously. Then, we test whether the network can solve the same task but with $\Delta t$ not included in their training set.

We find that in networks trained with multiple $\Delta t$, the single-neuron timescales $\tau$ follow a similar trajectory as the results in the main text, independent of $\Delta t$ (Fig. S5a,b compared to Fig. 4b). Multi-head networks adjust their $\tau$ to converge to $n\Delta t = 1$, while single-head networks increase their individual neuron timescale. Moreover, the networks can generalize (without retraining) the task to smaller $\Delta t$ than what was included in their training set, in single- and multi-head networks. Interestingly, the performance decreases slowly when $\Delta t$ becomes smaller than the training set, but abruptly when it becomes larger (Fig. S5c,d). The performance is best when training with multiple $\Delta t$, but qualitatively, the result is similar for a single, small enough $\Delta t$ (Fig. S5e).

## C  CHANGING THE DURATION OF THE INPUT PRESENTATION

In our tasks, the input contains two timescales. First is the duration of presentation of each input digit $T = k \cdot T_{\text{min}}$, with $T_{\text{min}}$ a minimal considered duration of stimulus presentation measured in milliseconds. Second is the timescale of the task's memory $N$. In the main text, we consider the situation of $k = 1$, but in general, $k$ acts as a time-rescaling parameter and defines one unit of time for the task performance. Here, we train the RNNs with different values of $k \in \{2, 3, 5, 10\}$ and check the trajectories of changing $\tau$ with $N$ depending on $k$. We find that similar to the case

with $k = 1$, single-head networks trained with $k > 1$ increase their $\tau$ with $N$, while multi-head networks try to keep $\tau$ close to $k$ (Fig. S6). Moreover, tasks with $k > 1$ are generally more difficult to solve since the input needs to be tracked over $N \cdot k$ time steps. Hence, as $k$ grows, RNNs would reach smaller $N$ within the same number of training epochs. The changes in values of $\tau$ after rescaling with $k$ might be due to nonlinear interactions in the network arising from the combination of different $N$ and $k$.

## D  PROPOSED ADDITIONAL TASK: TEMPORAL PATTERN GENERATION

For future research, the task variety can be extended to include the temporal pattern generation, which is a continuous-time task that is often used to evaluate RNNs (Durstewitz et al., 2023). The classic variation of the task involved an RNN receiving either random noise or no input and having to produce a target time series as output (usually a sum of sine waves with different frequencies).

A variation of the task we could consider for testing our model is the following:

**Single-head**: On the first step of the curriculum, we train the network to produce a single sine wave with frequency $f_1$, setting the target sequence to be $y_{N=1} = \sin(2\pi \cdot f_1 \cdot t)$.

Then, for each curriculum step, we complexify the target sequence by setting the new target as:

$$y_{N=m} = \sum_{i=1}^{m} \sin(2\pi \cdot f_i \cdot t), \tag{6}$$

for $f_1 > f_2 > \cdots > f_m$. In this way, as the newly added frequencies decrease, a need arises for the network to develop longer timescales.

**Multi-head**: Unlike the single-head network where the RNN needs to produce only one target time series $y_{N=m}$ at the $m$-th step of the curriculum, in the multi-head curriculum, the network produces $m$ output time series $Y = \{y_{N=1}, \ldots, y_{N=m}\}$.

## E  EFFECTS OF TRAINING WITHOUT A CURRICULUM ON THE $N$-DMS TASK

We investigate the negative effects of not using a curriculum during training for the $N$-DMS task to extend our results from Fig. 3a. We show in Fig. S13 that similar to the $N$-Parity task, networks rapidly lose the ability to solve the $N$-DMS task as $N$ increases when training without a curriculum. Interestingly, the two tasks differ in the way they fail to be solved despite using identical optimizers. In all of our results, the $N$-DMS task tends to be easier to solve for larger $N$. However, despite the relative success these networks have with the $N$-DMS task, their drop-off in training these networks is much steeper when comparing the curves from Fig. 3a and Fig. S13. In Fig. S13, tasks $N < 15$ get solved in only 1 or 2 epochs, however between $15 < N < 20$ the networks rapidly slow down in their ability to train until completely failing for $N > 20$ even when given longer training time. We can infer from these results that different tasks have varying degrees to which they benefit from a particular curriculum.

## F  INTERMEDIATE CURRICULA: MULTI-HEAD WITH A SLIDING WINDOW

The two curricula discussed in the main text (single-head and multi-head) represent two extreme cases. In the single-head curriculum, at each step of the curriculum, RNNs are trained to solve a new $N$ without requiring to remember the solution to the previous $N$s. On the other hand, in the multi-head curriculum, RNNs need to remember the solution to all the previous $N$s in addition to the new $N$. Here, we test the behavior of curricula that lie in between the two extreme cases.

The intermediate curricula involve the simultaneous training of multiple heads, similar to the multi-head curriculum, but instead of adding new heads at each curriculum step, we train a fixed number of heads and only shift the $N$s, which they are trained for according to a sliding window. We consider the number of heads to be 10, and start the training for $N \in [2, \ldots, 11]$. In the next steps of the curriculum, we use the already trained network to initialize another network which we train for $N + w$ (e.g., $N \in [2 + w, \ldots, 11 + w]$), where $w \in \{1, 3, 5\}$ indicates the size of the sliding

window. For each $w$, we train 4 different networks (i.e., 4 different initialization). For the following analyses, we trained the networks on the $N$-parity task.

We find that networks trained with the multi-head-sliding curriculum generally demonstrate an in-between behavior compared with the extreme curricula, but the results also depend on the size of the sliding window. Within 1000 training epochs, the maximal $N$ these networks can solve (with $> 98\%$ accuracy) is in between the maximal $N$ of single- and multi-head curricula, depending on the sliding window. Networks with a larger sliding window can solve a higher maximal $N$, indicating that a large sliding window not only does not slow down the training but also provides a more efficient curriculum to learn higher $N$s (Fig. S12a). Moreover, in multi-head-sliding networks, single-neuron ($\tau$) and network-mediated ($\tau_{\text{net}}$) timescales have values in between single-head and multi-head curricula (Fig. S12b). However, both $\tau$ and $\tau_{\text{net}}$ grow with $N$ similar to single-head networks, with the pace of growth reducing for larger sliding windows.

Similar to the main text (Fig. 7c,d,e), we perform the perturbation and retraining analysis on multi-head-sliding networks trained with $w = 5$. The relative accuracy after perturbation of recurrent weights $W^R$ and timescales $\tau$ for these networks lies between the two extremes (Fig. S14a, b). However, the retraining analysis suggests that multi-head-sliding networks can be retrained better for higher new $N$s (Fig. S14c,d). If the network is originally retrained for a small $N$ (e.g., $N = 16$), the retraining relative accuracy is similar between multi-head and sliding networks but is larger than single-head networks. For networks trained for larger $N$s (e.g., $N = 31$), sliding networks exhibit a superior retraining ability compared to the other two curricula. These results suggest that the curriculum with the sliding window helps multi-head networks to better adjust to new $N$s.

## G  SINGLE- AND MULTI-HEAD CURRICULA FOR TRAINING GRU AND LSTM

The results presented in the main text were generated using a modified version of a vanilla RNN (leaky-RNN) with an explicit definition of the timescale parameter $\tau$. To test whether the difficulties in training for long memory tasks without curriculum would carry over to recurrent networks that were specifically designed for long memory tasks, we train two other architectures, an LSTM (long short-term memory) and a GRU (gated recurrent unit) on the $N$-parity task for increasing $N$, with and without a curriculum. Both the GRU and LSTM have similar network sizes to the RNN with 500 neurons, though they differ in their activation functions (the RNN used a single leakyReLU whereas the GRU/LSTMs have both sigmoids and tanhs for different gates). Furthermore, in contrast with the RNNs, an Adam optimizer is used with learning rate $lr = 10^{-3}$ and the input signals to the models take values $\in \{-1, 1\}$ (to have a zero-mean input signal).

We find that for both architectures, training the networks without a curriculum is extremely slow for large $N$ and relatively unstable for small $N$ and probably requires strict hyper-parameter tuning (Fig. S15a). Without additional hyper-parameter tuning, introducing the multi-head curriculum speeds up the training significantly, and both architectures can easily learn the $N$-parity task with large $N$ similar to the leaky-RNN (Fig. S15b). Moreover, similar to RNNs, the multi-head curriculum has a higher training speed than the single-head curriculum (Fig. S16). Our results indicate that GRUs and LSTMs are subject to similar training dynamics as RNNs used in the main text and the multi-head curriculum is an optimal curriculum regardless of the RNN architecture. The advantage of using the leaky-RNN architecture is that its parameters are easier to interpret, and it allows us to study better the mechanisms underlying each curriculum by explicitly studying the role of timescales.

## H  BACKWARD AND FORWARD RETRAINING OF NETWORKS

To understand how trained models develop their ability to create longer timescales throughout the curriculum as well as their backward compatibility and robustness to catastrophic forgetting, we measure the retrainability of models trained on a task with memory $N$ on a different task with memory $N^*$. We freeze all parameters of a trained network except the final readout layer weights which are retrained on an $N^*$ task. Specifically, we load models trained for $N \in [2, \dots, 19]$ and retrain them on a new $N^* \in [2, \dots, N+2]$, independently for each $N^*$, for a maximum of 10 epochs or until its accuracy was above $98\%$. Note that we retrain both single- and multi-head networks as single-head.

We find that the multi-head networks exhibit near-perfect backward compatibility as well as better forward compatibility than the single-head models (Fig. S17), while single-head networks suffer from catastrophic forgetting. For the multi-head networks, the backward compatibility is enforced through the loss function (as is the case in the multi-head curriculum) hence, the necessary representations for $N^* < N$ persist. However, the multi-head curriculum also has positive implications for forward compatibility, which is evident in the off-diagonal entries of the accuracy where $N^* > N$ (to the right of the dotted line) when compared to the single-head values.

## I EMERGENCE OF CURRICULUM DURING MULTI-HEAD TRAINING

In the multi-head curriculum, the difficulty of the task increases gradually; a new head with a larger $N$ is added at each step of the curriculum. In the main text, we discussed that networks trained with such a curriculum generally train well up to large $N$s. Here we ask whether this optimal curriculum can emerge by itself if we train a network with multiple heads, but without any predefined curricula. For this analysis, we train RNNs with 19 ($N \in [2, \ldots, 20]$) and 39 heads ($N \in [2, \ldots, 40]$) to solve all the available $N$s simultaneously.

We find that despite the absence of an explicit curriculum, these networks learn the task by generating an internal multi-head curriculum. While all the heads contribute equally to the loss, heads with a smaller $N$ reach the higher accuracy faster (Fig. S18a). However, the speed of training strongly depends on the total number of heads in each network. For the same $N$, the network with 19 heads reaches the 98% accuracy faster than the network with 39 heads (Fig. S18b), but both networks have a slower training speed when compared to the multi-head curriculum. These results suggest that the multi-head curriculum is an optimal curriculum that can arise naturally during multi-head training and can increase the training speed when applied explicitly.

## J ROLE OF STRONG INHIBITORY CONNECTIVITY IN SINGLE-HEAD NETWORKS

The main difference between single- and multi-head networks in terms of connectivity is the stronger inhibitory (negative) connectivity for large $N$ in the single-head networks compared with the relatively balanced connectivity in multi-head networks (Fig. 6a). We hypothesized that larger inhibition in single-head networks is required to keep the dynamics stable in the presence of slow single-neuron timescales $\tau$. To test this hypothesis, we perturb only the inhibitory connections in networks trained with both curricula as:

$$W_{ij}^R = W_{ij} + c \cdot W_{ij}, \qquad \forall W_{ij}^R < 0, \tag{7}$$

for a given amount of $c \in [-0.1, 0.1]$. We observe that by reducing the amount of inhibition in single-head networks, the network activity explodes even before reaching the balanced point, i.e., the point when the average incoming weight of neurons becomes 0 (Fig. S19a). On the contrary, multi-head networks are significantly more robust to such perturbations and their activity remains within a reasonable range for a broad range of inhibitory scaling (Fig. S19b).

This difference is most likely attributed to the difference in single-neuron timescales $\tau$ between single- and multi-head networks. The single-head networks have a larger average $\tau$ compared to the multi-head networks whose average $\tau \approx 1$ for large $N$. Longer $\tau$ leads to neurons with self-sustaining activity, and thus, a stronger inhibition might be required to prevent the runaway activation. Such a relationship can be observed when comparing the average $\tau$ and inhibitory strength across networks: for single-head networks as $\tau$ grows, the average weight becomes more negative (inhibitory)(Fig. S19c), but such correlation does not exist in multi-head networks (Fig. S19d).

## K DEPENDENCE OF DIMENSIONALITY OF POPULATION ACTIVITY ON $N$

We measure the dimensionality as the number of principal components that explain $90\%$ of the population activity variance. The dimensionality increases with $N$ for both tasks and curricula, but the increase follows a linear relation with $N$ for $N$-parity task but a sub; linear relation for the $N$-DMS task (Fig. 6b).

To demonstrate this difference, we fit two separate lines for the data up to $N = 20$ and from $N = 20$ up to the largest $N$. We observe that for the $N$-parity task, the slope of two lines largely overlaps, indicating a linear relation. However, for the $N$-DMS task, the second line clearly has a smaller slope than the first one, indicating a sub-linear growth with $N$ (Fig. S20).

## L    ABLATION DETAILS

To test whether neurons with fast or slow timescales ($\tau$) are necessary for computations in the trained RNNs we perform the ablation analysis. For this analysis, we compute the relative accuracy of the model (Eq. 4 in the main text) after removing a single neuron. We ablate neuron $i$ by setting all incoming and outgoing associated weights to zero

$$
\begin{aligned}
W_{ij}^{R}, W_{ji}^{R} &= 0 \qquad \forall j \\
W_{i}^{O}, W_{i}^{I} &= 0
\end{aligned}
\tag{8}
$$

Here $W_{ij}$ refers to recurrent weights, $W_{i}^{O}$ to input weights and $W_{i}^{O}$ to readout weights. To measure the relative accuracy, we simulate the RNN forward using random binary inputs for 1000 time steps after 100 time steps of a burn-in period (to reach the stationary state). Then, we evaluate the accuracy of the network at each time step. We repeat this procedure over 10 trials and compute the average and standard deviation of the relative accuracies across trials.

## M    SIGNIFICANCE OF THE RESPONSES TO PERTURBATIONS OF WEIGHTS AND RETRAINING

We investigate the significance of differences between single- and multi-head networks presented in Fig. 7 using a t-test (two-sided, unpaired). Perturbations are computed 10 times for 4 networks per group with results being pooled across networks. Retraining accuracy is computed once per network. Table 1,2, and 3 indicates with stars the significance levels corresponding to p-values below $5e-2, 1e-2, 1e-3, 1e-4$, and $1e-5$.

| Weight Perturbation Strength | p-value | Significance |
|:---:|:---:|:---:|
| 1.0e-02 | 8.8e-03 | ** |
| 2.2e-02 | 4.5e-01 | n/s |
| 4.6e-02 | 2.9e-01 | n/s |
| 1.0e-01 | 5.1e-15 | ***** |
| 2.2e-01 | 2.2e-39 | ***** |
| 4.6e-01 | 8.3e-47 | ***** |
| 1.0e+00 | 8.6e-04 | *** |
| 2.2e+00 | 7.6e-01 | n/s |
| 4.6e+00 | 6.5e-01 | n/s |
| 1.0e+01 | 9.0e-01 | n/s |

Table 1:   Significance of the weights' perturbation for different perturbation sizes Fig. 7c. Two-sided and unpaired t-test, stars indicate p-values below $5e-2, 1e-2, 1e-3, 1e-4$, and $1e-5$.

| Perturbation of $\tau$ | p-value | Significance |
|---|---|---|
| 1.0e-03 | 4.3e-01 | n/s |
| 2.2e-03 | 8.9e-01 | n/s |
| 4.6e-03 | 5.4e-01 | n/s |
| 1.0e-02 | 1.2e-02 | * |
| 2.2e-02 | 2.3e-14 | ***** |
| 4.6e-02 | 9.1e-29 | ***** |
| 1.0e-01 | 1.3e-50 | ***** |
| 2.2e-01 | 4.1e-35 | ***** |
| 4.6e-01 | 4.8e-01 | n/s |
| 1.0e+00 | 6.0e-01 | n/s |
| 2.2e+00 | 1.3e-01 | n/s |

Table 2: Significance of the $\tau$'s perturbation for different perturbation sizes Fig. 7d. Two-sided and unpaired t-test, stars indicate p-values below $5e-2, 1e-2, 1e-3, 1e-4$, and $1e-5$.

| retraining for $N$ | p-value | Significance |
|---|---|---|
| 17 | 1.7e-03 | ** |
| 18 | 9.7e-04 | *** |
| 19 | 1.2e-07 | ***** |
| 20 | 3.2e-05 | **** |
| 21 | 1.7e-05 | **** |
| 22 | 1.6e-05 | **** |
| 23 | 7.4e-06 | ***** |
| 24 | 6.4e-05 | **** |
| 25 | 3.1e-04 | *** |
| 26 | 1.7e-03 | ** |
| 27 | 1.7e-02 | * |
| 28 | 2.1e-03 | ** |
| 29 | 7.5e-03 | ** |
| 30 | 1.6e-01 | n/s |
| 31 | 1.5e-01 | n/s |
| 32 | 6.5e-01 | n/s |

Table 3: Significance of the retraining differences between single and multi-head, Fig. 7e. Two-sided and unpaired t-test, stars indicate p-values below $5e-2, 1e-2, 1e-3, 1e-4$, and $1e-5$.

## N  CODE AND DATA AVAILABILITY

Codes for training and evaluating the RNNs and reproducing the experiments (e.g., measuring timescales, performing ablations, etc.) together with example trained networks are available on GitHub at https://github.com/LevinaLab/rnn_timescale_public (more details in README).

# O SUPPLEMENTARY FIGURES

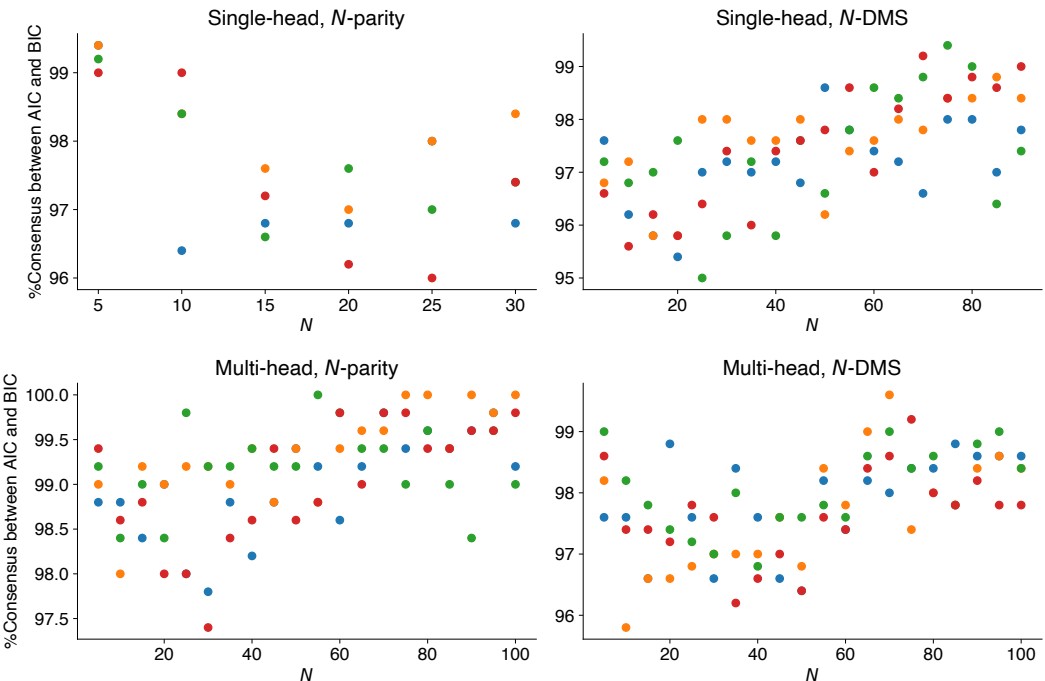

Figure S1: AIC and BIC choose the same model for most neurons. We fitted AC of each neuron's activity with single- and double-exponential functions and used AIC or BIC to select the best-fitting models. The results show that for above 95% of neurons, the two criteria select the same model. The colors of the dots indicate different networks (4 networks for each task, curriculum and $N$).

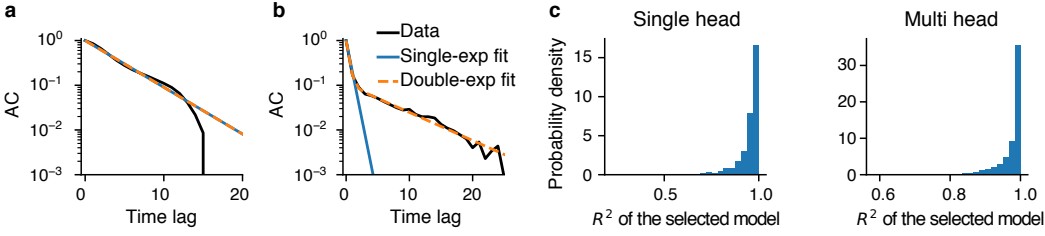

Figure S2: ACs of neurons are well captured with single- or double-exponential fits. Note that the y-axis is in logarithmic coordinates, meaning that deviations between the fit and data AC are much smaller in the AC tail compared to initial time lags. **a, b.** Fitting double and single exponential functions to the AC of example (a) single-timescale ($\tau_{\text{net}} = \tau$) and (b) double timescale ($\tau_{\text{net}} > \tau$) neurons. **c.** Values of coefficient of determination $R^2$ estimated for all selected fits using AIC are close to 1, indicating a good fit.

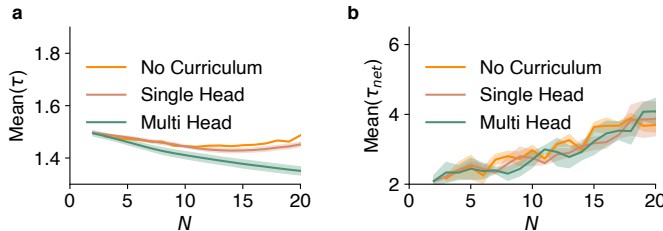

Figure S3: Networks trained without curriculum have similar single-neuron (**a**) and network-mediated (**b**) timescales to networks trained with the single-head curriculum in the range of N that the no-curriculum-trained networks can learn.

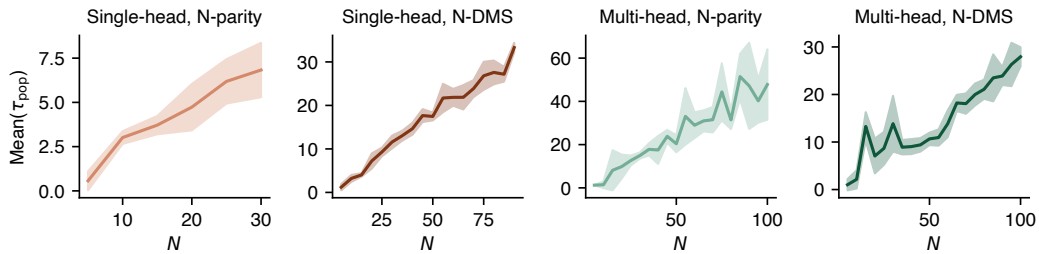

Figure S4: Dependence of population activity timescales $\tau_{\text{pop}}$ on $N$. For both tasks and curriculum, the timescale of population activity fluctuations increases with $N$, indicating a general trend toward slower collective dynamics for tasks with larger memory requirements. Shade - $\pm$ STD across 4 networks.

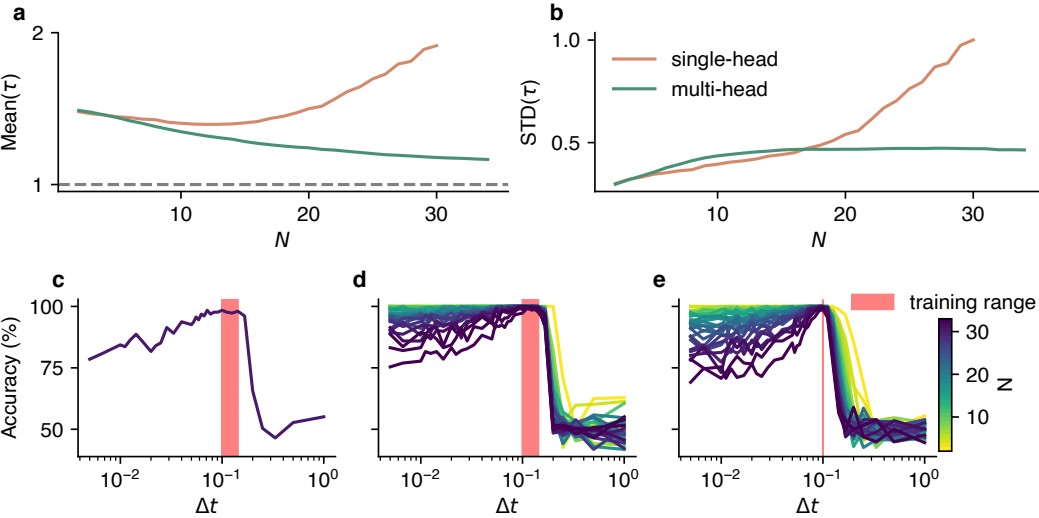

Figure S5: Impact of discretization time-step $\Delta t$ on training performance. We train networks with $\Delta t = \{\frac{1}{10}, \frac{1}{9}, \frac{1}{8}, \frac{1}{7}\}$, while presenting each input digit for the duration of $T = 1$. (**a**) Similar to discrete-time networks ($\Delta t = 1$), the mean of single-neuron timescales $\tau$ increases with $N$ for single-head networks and decreases towards $T = 1$ for multi-head networks. (**b**) The standard deviation of $\tau$ indicates heterogeneous $\tau$s for single-head networks but constrained values for multi-head networks. (**c,d**) Single-head (c) and multi-head (d) networks can solve the task above the chance level for $\Delta t$ smaller than their training regime (indicated by the red rectangle) when trained with multiple $\Delta t$. (**e**) The networks are slightly more inaccurate when trained on only a single $\Delta t = \frac{1}{10}$. Lines and the color bar indicate different $N$.

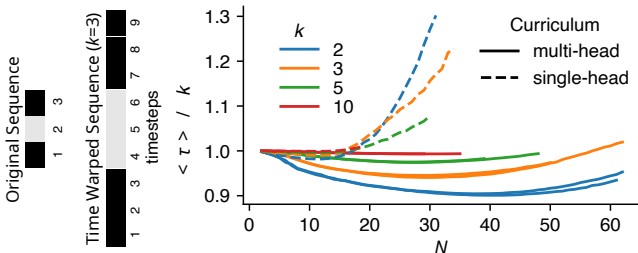

Figure S6: Changing the duration of input presentation. Each input digit is presented to RNN for a duration of $T = k\Delta t$, $\Delta = 1$. Single-neuron timescales ($\tau$s) normalized by $k$ remain roughly constant in multi-head networks (i.e. $\tau \to k\Delta t$), but increase with $N$ in single-head networks (cf. Fig. 4b).

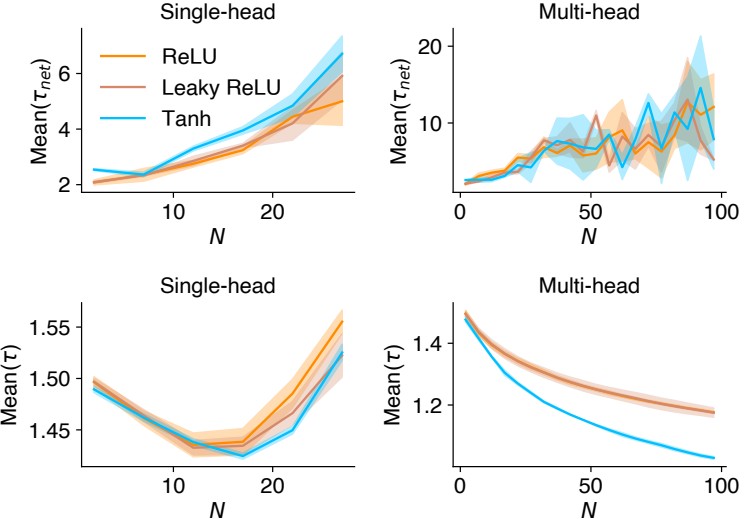

Figure S7: The development of timescales follows similar trajectories when the self-interaction is inside (nonlinear $\tau$) or outside (linear $\tau$) the nonlinearity (leaky-ReLU). Top: network-mediated timescales, bottom: single-neuron timescales. Shades - $\pm$ STD.

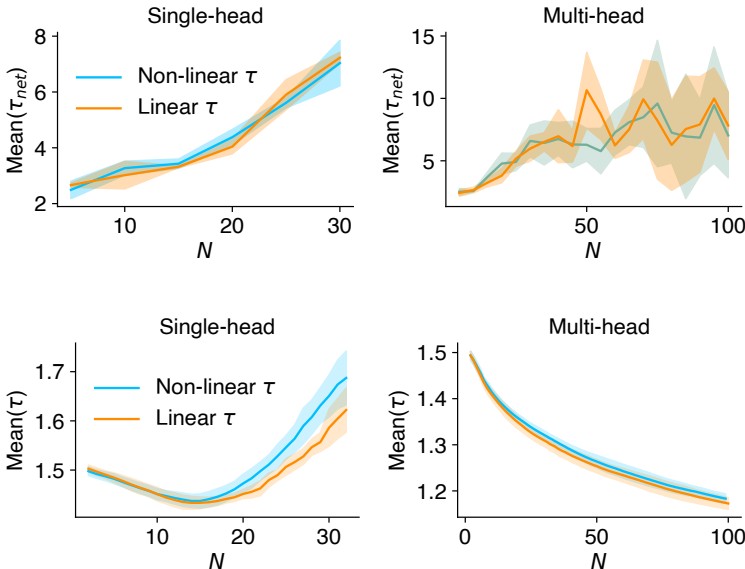

Figure S8: The development of timescales in networks with different nonlinearities follows similar trajectories. Top: network-mediated timescales, bottom: single-neuron timescales. Shades - $\pm$ STD.

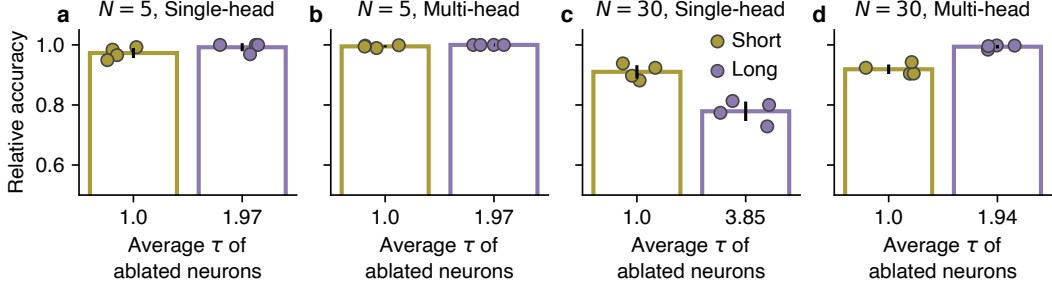

Figure S9: Impact of ablating neurons with distinct timescales on RNNs' performance when neural self-interactions are linear (cf. Fig. 7a,b). **a, b.** Ablating the longest and shortest timescale neurons has minimal effect on network performance when $N$ is small for both curricula. **c, d.** For higher $N$, ablating long timescale neurons largely decreases the performance of single-head networks, while multi-head networks are more affected by the ablation of short-timescale neurons. Bars - mean, error bars - STD, dots - 4 individual networks.

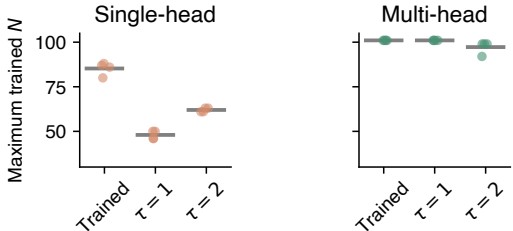

Figure S10: The maximum $N$ solved in the $N$-DMS task after 1000 epochs (reaching an accuracy of 98%). Similar to the $N$-parity task (cf. Fig. 4a), models trained with a single-head curriculum rely more on training $\tau$ than the multi-head curriculum networks, which prefer to have a small $\tau$ and are more agnostic to it being trainable. Horizontal bars - mean.

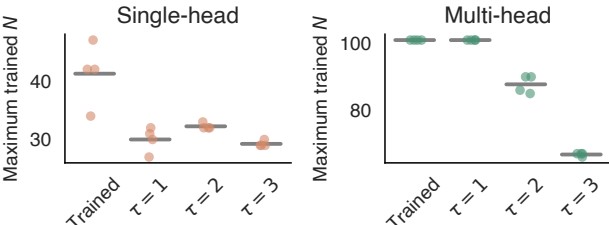

Figure S11: The maximum $N$ solved in the $N$-parity task (for models with the self-interaction outside the nonlinearity) after 1000 epochs (reaching an accuracy of 98%). In the single-head curriculum, models rely on training $\tau$, whereas in the multi-head curriculum, having $\tau$s fixed at 1 value is as good as training them. Results are consistent with the models where the self-interaction is inside the nonlinearity (cf. Fig. 4), Horizontal bars - mean.

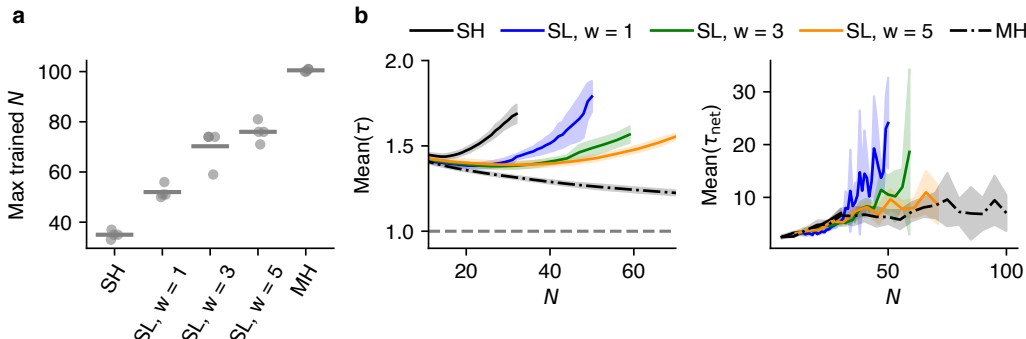

Figure S12: The behavior of networks trained with multi-head-sliding curriculum depends on the size of the sliding window and lies in between extreme curricula. **a.** The maximal trained $N$ (with $> 98\%$ accuracy, within 1000 training epochs) for multi-head-sliding (SL) lies between single-head (SH) and multi-head (MH) networks and increases with the size of sliding window ($w$). Dots indicate individual networks (4 networks) and the horizontal bars indicate the mean value. **b.** Single-neuron ($\tau$) and network-mediated ($\tau_{\text{net}}$) timescales increase with $N$, but the pace of change reduces as the sliding window grows. Shadings indicate $\pm$ std computed across 4 trained networks.

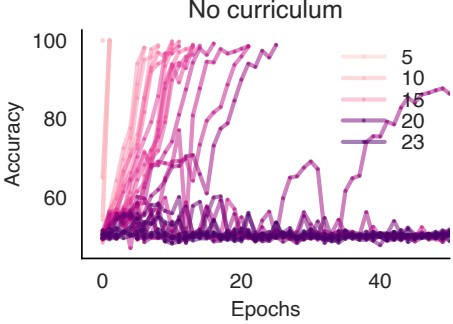

Figure S13: Training without a curriculum on the $N$-DMS task. For each $N$, 4 models are independently trained for 50 epochs or until reaching $> 98\%$ accuracy. Similar to the $N$-Parity task (cf. Fig. 3a), the ability to solve the task decreases as we increase N. From $N > 20$, we see the network is no longer capable of finding a solution to the task even with longer training time.

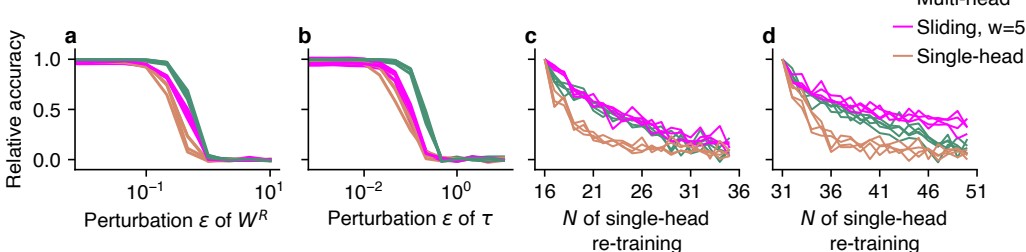

Figure S14: Robustness of networks trained with multi-head-sliding curriculum. **a, b.** Multi-head-sliding networks are more robust than single-head networks but less robust than the multi-head networks against perturbations of recurrent connectivity (a) and trained timescale $\tau$ (b). Each line indicates one trained network (4 networks for each curriculum). Shades indicate $\pm$ std computed across 10 trials. **c, d.** retraining of networks trained with different curricula as a single-head network on new $N$s (for 20 epochs). Multi-head-sliding networks achieve higher relative accuracy when retrained for a higher $N$ in comparison to single-head networks. If originally trained for small $N$s (c, $N = 16$), they have similar retraining accuracy to multi-head networks, but for larger $N$s (d, $N = 31$) their accuracy suppresses the multi-head networks. Each line indicates the relative accuracy for one network (4 networks for each curriculum).

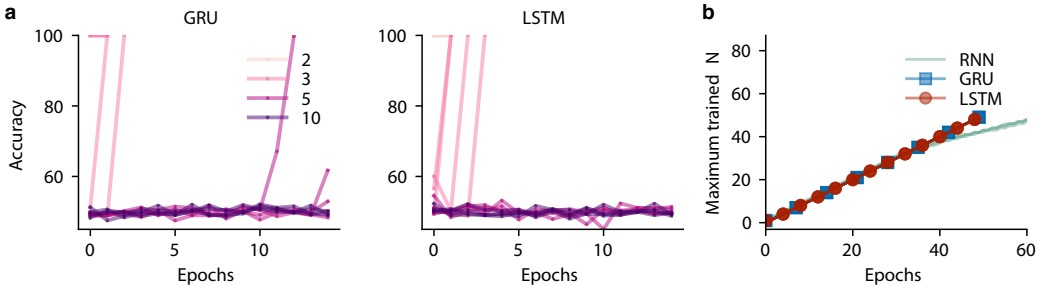

Figure S15: Comparing the impact of curriculum on different recurrent architectures. Two different architectures, GRU and LSTM, are trained on the $N$-Parity task with and without a curriculum. We observe that the GRU and LSTM both exhibit instability when training without a curriculum (**a**), but are comparable to the RNNs with the multi-head curriculum (**b**).

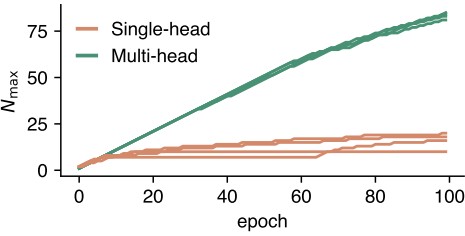

Figure S16: Comparison of single- and multi-head curricula for training LSTMs on $N$-parity task. Networks trained with the multi-head curriculum can reach a higher $N$ faster than networks trained with the single-head curriculum.

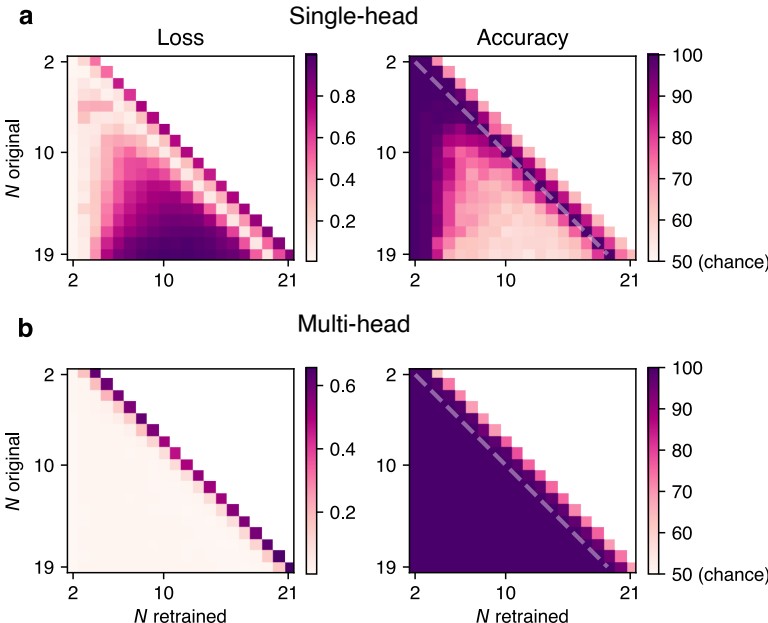

Figure S17: Single-head (a) and multi-head (b) networks loaded for $N \in [2, \ldots, 19]$ have new readout heads retrained on new tasks with $N^* \in [2, \ldots, N + 2]$. The heat map of the loss and accuracy of these retrained networks (after a maximum of 10 epochs or reaching an accuracy of 98%+) shows the robustness of the multi-head networks to catastrophic forgetting, as well as an improvement towards forward compatibility in the $N^* > N$ region. The dotted line indicates the diagonal.

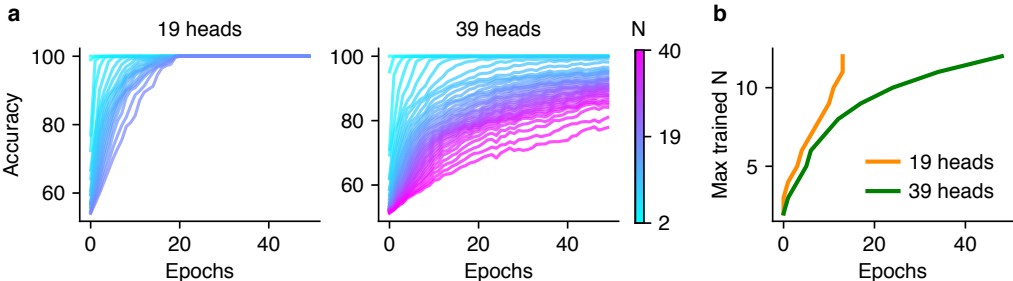

Figure S18: Emergence of curriculum during multi-head training. **a.** In the absence of an explicit curriculum, multi-head networks solve smaller $N$s before solving the large $N$s. The color bar indicates the range of $N$s. **b.** The speed of training reduces with the increasing number of heads. The network with 19 heads needs fewer epochs to solve the same $N$ (i.e. reaching 98% accuracy) than the network with 39 heads.

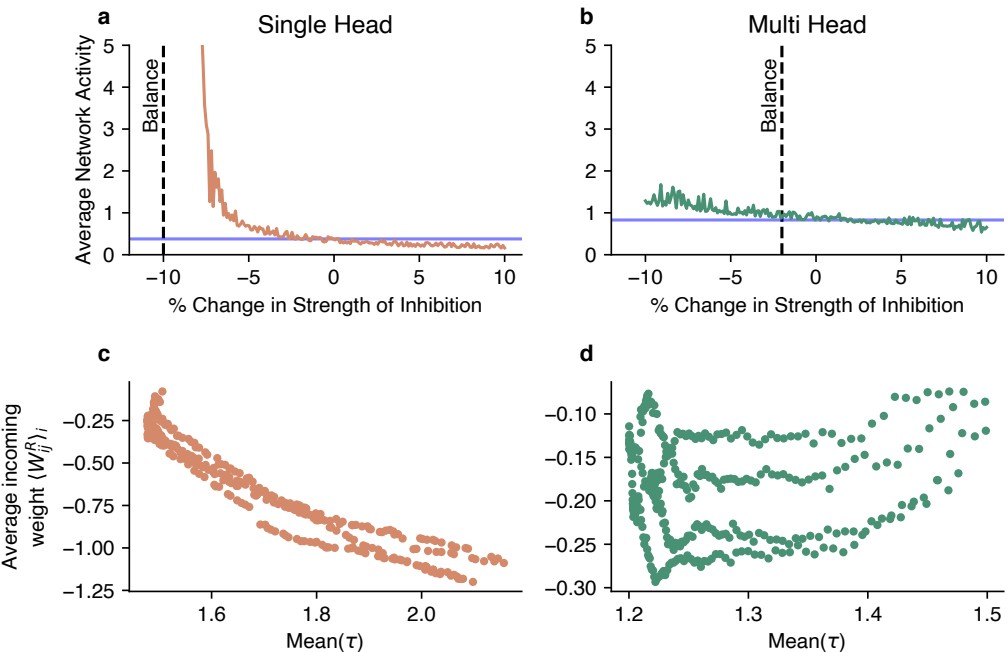

Figure S19: Networks trained with the single-head curriculum require strong inhibitory connectivity. We perturb the inhibitory (negative) connections of a single (**a**) and a multi-head network (**b**). We see that the activity of the single-head network explodes as we approach the balanced point (the point where the average of incoming weights becomes 0, indicated by the horizontal blue line). On the contrary, the multi-head network is quite robust and produces activity within a normal range even after the balanced point. (**c**) In the single-head networks, we observe a negative correlation between the average $\tau$ and the average strength of incoming weights for each neuron (i.e. higher $\tau$ is correlated with more negative average weight). This relationship is not present for multi-head networks (**d**) that are largely balanced and maintain small $\tau$.

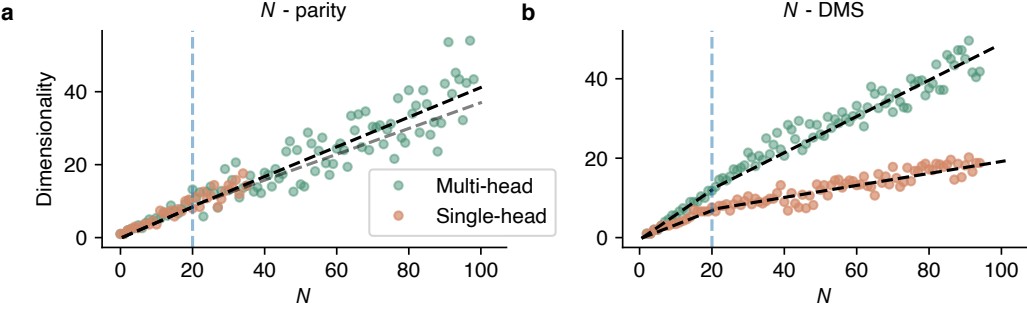

Figure S20: Dimensionality of activity increases approximately linearly with $N$ for the $N$-parity task and sub-linearly for the $N$-DMS task. We separately fit the data points for $N \in [0, \dots, 20]$ and $N \in [20, \dots, 100]$ ($N \in [20, \dots, 30]$ for the single-head $N$-parity network) and we observe that in the $N$-parity task, the two lines largely coincide, while in the $N$-DMS case there is a clear change in the slope of the line, suggesting a sub-linear increase with $N$.

