# OpenReview forum: "Emergent mechanisms for long timescales depend on training curriculum and affect performance in memory tasks"
_ICLR.cc/2024/Conference — ICLR 2024 poster_

### Official Review · Reviewer_TkDB · 2023-10-17

**Soundness:** 3 good
**Presentation:** 3 good
**Contribution:** 3 good
**Rating:** 8
**Confidence:** 3

**Summary:**

The authors explore how RNNs derive a solution during curriculum learning for working memory tasks. In particular, in pursuit of functional working memory the authors contrast the optimisation of intrinsic neuronal timescales, as seemingly favoured by single-headed RNNs which are trained on one task, with the optimisation of recurrent weights, as favoured by multi-headed RNNs which are trained on several tasks at once.

**Strengths:**

- The paper is generally well written
- The authors discover, in my view, an interesting and convincing contrast in the solutions obtained by single vs multi-head learning with respect to intrinsinc vs network mediated timescales.
- I appreciate the depth of analysis undertaken by the authors with respect to the performance and dynamics of the networks, and generalisability to other network architectures

**Weaknesses:**

- I think this paper could and should be valuable to neuroscientists, but at the moment the link between the paper results and biology is not so clear. For example, do the authors surmise that optimised, heteregenous instrinsic neuronal timescales should not be employed in the brain? Would there be any experimental predictions one could infer from this work?
- Related to the above, it is not clear to me if other potential enablers of short-term memory have been considered. For example, there is one experimental work (Hu et al. 2021, PLOS) which comes to mind which shows that short-term plasticity (STP) in neurons, rather than recurrent connectivity, is the more consistent enabler of short term memory in a visual task. Similarly, one computational work (Salaj et al. 2021, Elife) promotes spike frequency adaptation as a powerful tool for sequence processing. I would like to know the authors' opinion on these additional neural mechanisms known in biological circuits.
- The tasks selected are relatively simple and it is not clear if these results generalise to harder temporal tasks, though this is addressed as a limitation of the study. More generally, I would caution the authors' with the implication that their findings can generalise to "networks on sets of related tasks instead of a single task".The authors do demonstrate their case well for the setting of learning different N simultaneously during curriculum learning, but whether these results apply when, e.g. language processing in two different languages, is not clear to me.
- The motivation for two working memory tasks: N-parity and N-DMS, is not very well presented. What are the differences between the tasks. The authors note that Fig 6b reflects "distinct computational requirements for each task" but this is not elaborated upon.

**Questions:**

- In the 4th paragraph of section 2 it is written tau_{net} <= tau, should it not be the other way around?
- For fig 3a, which task is that? N-parity? The learning curves on this plot also appear to be still going up; it is not also clear whether these N will eventually be learned.
- Section 3.1: "N -parity: The network has to output the binary sum (XOR) of the last N digits, indicating an odd or even number of non-zero digits". I don't understand this; is it that the model should output whether the sum of the last N digits is odd or even?
- Same N-Parity section: "the value of the digit presented at t − N needs to be subtracted from the running sum". I am not such a fan of this description as the authors' are suggesting the underlying mechanism for how the RNN solves the problem is known. Perhaps the RNN is incorporating another solution; e.g. perhaps N running sums are simultaneously encoded in the RNN and are reset every N timesteps.
- For Figs 3a, 4a, do these results also apply for the N-DMS task? Why are only
- is k > 1 ever considered? for the multi-head case the authors claim that tau -> k, and this seems true for k = 1; was this also tested for higher k?
- For Fig 3 was tau fixed? so only recurrent connectivity optimised? This should be made clear
- is Fig 5a for the single-headed network?
- For Fig 5d in each task the network timescale tau_net increases up to about N=30 then shrinks. Is this shrink surprising? It is also unclear to me how these multi-headed network can perform the task so well, when their tau_net is overall considerably smaller (about half the size) of the worse-performing single-head networks
- Fig 6b: I was surprised at how high dimensional these network activities are. In my experience trained RNNs produce quite low dimensional dynamics (< 10 or often < 3). Is this high dimensionality surprising to the authors?
-

Typos and suggestions:
- add (AC) after autocorrelation in the Fig 1 caption.
- In the 4th paragraph of section 2: tnet -> tau_{net}
- I would appreciate a more formal, mathematical formulation of how tau_{net} was obtained for each neuron in the appendix
- End of first paragraph of section 4.1, remove comma after K
- I'd recommend the related work prior to the results
- I'd recommend all supplementary figures together at the very end. Their appearance amongst the different appendix sections seems quite random/distracting

---

> ### Author Response · Authors · 2023-11-20
>
> We thank the referee for constructive comments, especially the suggestion of highlighting the relevance of our work for the neuroscience community. We addressed raised concerns by performing new analyses and improving the text, as we describe below (submitted as two comments).
>
> - Relevance to neuroscience:
>
> Thank you for recognizing the neuroscientific interest in our results; we are primarily motivated by insights from and for neuroscience. To make this more clear, we added a new section 5 on the neuroscientific implications of our findings.
>
> - Single neuron vs network timescale:
>
> We agree with the referee; this is a fascinating point, and we should expand on it and highlight the possible biological implications. Our results indicate that adapting single-neuron timescales to the task requirements is less robust than learning based on adjustment of synaptic weights. This is greatly in line with how the neural networks in the brain primarily train: although there are possibilities to adjust single-neuron timescales, primary learning is implemented in adjusting the connectivity.  However, the heterogeneity in single-neuron timescales might be very important in the brain and our networks. We changed the duration of each stimulus presentation and found that the individual timescales track the input timescale in single and multi-head networks (Appendix C). This proposes one possible mechanism for how single-neuron timescales (equivalent to the membrane constant in our formalism) might be important for computations in reality, where the input timescales could vary a lot, e.g., different input timescales might be tracked by neurons with different taus.
>
>
> - Short-term plasticity and adaptation:
>
> Indeed, STD/STP and adaptation are important in biological networks. Here, we simplify the biological picture for the sake of having only two dominating timescales, however, the inclusion of such mechanisms would introduce additional timescales generated by the plasticity/adaptation. From our results, we would expect that matching these timescales to the input could, for example, change which tau’s are optimal. In biological networks with given adaptation or STD/STP mechanisms, we believe that the learned recurrent connectivity will still be the defining factor for the resulting timescales, but expect more complicated interaction between different mechanisms. Moreover, previous work suggests that single-neuron adaptation timescale (Beiran & Ostojic, PLoS CB, 2019) does not affect collective network timescales that are probably more relevant for computations. We added this discussion to the limitations section and included the suggested references.
>
> - Task choice and generalization:
>
> We agree with the referee and weakened our statements and highlighted this in limitations.
>
> - N-DMS and parity:
>
> The difference between these two tasks is that N-DMS only requires memory of input, whereas N-parity also needs computation (though quite simple). We add this briefly to the text (section 3.1).

---

> > ### Author Response · Authors · 2023-11-20
> >
> > Questions:
> >
> > - Typo in section 2: Thank you, we corrected it.
> >
> > - Results for N-DMS task (Fig.3a ,4a) and Learning curves:
> >
> > Figure 3a. is for the N-Parity task (we now added to the caption). We have also made an equivalent figure for the N-DMS task (Appendix E, Fig. S13). These figures are meant to demonstrate how the rate of learning decreases as the task complexity (N) increases. In the case of N-DMS, we can explicitly see it fails to solve the task for N > 20 (Fig. S13) and for N-parity, this is also true for slightly larger N, though we only included up to N=20 in Fig. 3a. Furthermore, this figure illustrates how slow the learning in this case is, particularly compared to a multi-head network that reaches N=50 at 50 epochs.
> >
> > An equivalent figure for Fig 4a. is added for the N-DMS task (Fig. S10) for values of tau fixed at 1 and 2. We observe similar patterns as with the N-Parity, but with smaller differences, most likely due to a simpler task.
> >
> >
> > - Description of the tasks, particularly N-parity:
> >
> >  Thanks for pointing out the problem with our description. Our aim with that description was to clarify that since the computations are online (the network needs to produce an output every time step), it needs somehow to keep a memory of the last N digits, but, indeed, we do not know the exact mechanism for solving the task. We have edited the text accordingly.
> >
> > - 4a: Results for N-parity and N-DMS:
> >
> > Apologies for the misleading k-notation; and added a more detailed explanation about k interpretation in new section 5 and Appendix C.
> >
> > - Fig. 3: fixed timescales:
> >
> > The timescales are fixed only in Fig. 4a (and corresponding Fig. S10).
> >
> > - Figure 5. Meaning of stalling the timescales:
> >
> > Panel a is for single-head networks (now added to the caption). Our interpretation for stabilization of tau_net is that, for large N, networks need to find solutions other than increasing the timescales of local outputs (measured by tau_net). They are possibly employing combinations of neurons that can have different timescales on the population level than individual neurons if they are well-coordinated. Since we cannot assess all the combinations of local outputs' contributions to the readout, we designed a proxy. We took the whole population and summed up the activity of all neurons, obtaining the timescale of the population activity tau_pop. For all tasks and training curricula, tau_pop grows with N (Fig. S4).
> >
> > - High dimensionality of the outputs compared to other RNN studies:
> >
> >  It is indeed surprising when we compare the dimensionality to other neuro studies with RNNs. Part of the reason for this can be that the common neuro tasks are by design low-D. For instance, tasks usually have a fixed trial structure, and the low-D trajectories often follow the events within a trial. For example, a DMS task is often designed to start with one stimulus, followed by the delay and then the 2nd stimulus. The only variable is the delay period. However, in our case, the computations have to be performed at every time step, and very different input patterns (the number patterns grow with N) should be mapped to the same outputs (0,1). Nevertheless, it is a point for further investigation.
> >
> > - Typos and suggestions:
> >
> > Thanks for noting these. We corrected the typos and edited the text.
> >
> > We moved all supplementary figures to the end. However, we did not change the location of the Related Work section since it includes a comparison to our findings; instead, we edited it to better reflect these comparisons.
> >
> > Moreover, since the definition of tau_net is an important part of our paper, we edited the definition in the main text to provide more detailed explanations.

---

> > > ### Comment · Reviewer_TkDB · 2023-11-22
> > >
> > > Thank you to the authors for their detailed rebuttal as well as the significant changes and extensions they have made to the manuscript.
> > >
> > > I believe the work is stronger for it and believe this paper to make an important contribution both for machine learning and neuroscience. I will update my score accordingly.

---

> > > > ### Author Response · Authors · 2023-11-22
> > > >
> > > > Thank you for considering our rebuttal and updating the score! We really appreciate your recognition of our work and your feedback.

---

### Official Review · Reviewer_XC2K · 2023-10-17

**Soundness:** 3 good
**Presentation:** 3 good
**Contribution:** 3 good
**Rating:** 8
**Confidence:** 3

**Summary:**

Training RNNs can adjust the time scales of their responses in order to perform memory tasks. The authors point out that this may involve adjustment of the time scales of individual neurons, or of the network-generated timescales created by interactions between neurons. They set out to tease apart the roles of either component in different memory tasks and different training settings.

They train RNNs on two memory tasks whose difficulty increases with sequence length N (N-DMS and N-parity), and under two settings (successive Ns on a single network, and all-N-so-far with different heads for different Ns).

 They show that the single-head setting tends to favor changes in single-neuron timescale, while the multi-head setting tends to keep single-neuron timescale low and increases network-generated timescales instead.

Various ablations and analyses confirm this observation. The multi-head networks are shown to rely more on shorter-timescale neurons, while the single-head networks rely more on the longer-timescale neurons (whose tiomescales are much longer in the first place). Multi-head networks are more robust to parameter perturbation.

**Strengths:**

- I'm not aware of this question (whether RNNs learn more by adjusting single-neuron or network-generated timescales) having been addressed previously.

- The analyses are convincing, though some clarifications are needed.

**Weaknesses:**

I did not see any obvious "weakness" in the demonstration. I do note that the tasks and settings are quite specific, and it's not clear how instructive the results are for more realistic domains (this is also noted by the authors in the Limitations)

**Questions:**

- Most glaringly: The authors themselves point out that tau=1 makes the network memoryless. Yet in Figure 4 they show the networks easily solve difficult memory tasks with tau fixed at 1? How is that even possible?

- The initial text suggests that tau and tau_net are independent components (one strictly single-neuron, one strictly network-generated). However, later on it is acknowledged that longer tau causes longer tau_net. Some clarification would be in order.

---

> ### Author Response · Authors · 2023-11-20
>
> We thank the referee for a thorough review and appreciate the positive feedback and finding our results promising. We address the questions raised below.
>
> - Memorylessness at tau=1:
>
> $\tau = 1$ makes individual neurons memoryless in the absence of any network interactions. However, the network as a whole is not memoryless, and recurrent inputs to each neuron create correlations between the activity of neurons across different time steps (as one example, you can imagine the wave of activity propagating through the network in circles). That means when $\tau = 1$, the memory capacity of the network is fully determined by the connectivity. We now clarify this in the text (pages 2-3).
>
> - Relationship between $\tau$ and $\tau_{net}$
>
> $\tau_{net}$ is generally a function of tau and the recurrent weights. We edited the text and provided a better explanation (page 3).

---

> > ### Comment · Reviewer_XC2K · 2023-11-22
> >
> > I have left my evaluation unchanged.
> >
> > With respect to remarks by other reviewers, I am not aware of much existing work concerning the *specific* question tackled in this paper, namely, the distinct single-neuron vs network-generated timescales, and the factors that might affect either and their relative importance. I would be happy to learn more about any such work, if there is any.

---

### Official Review · Reviewer_h2ki · 2023-11-02

**Soundness:** 3 good
**Presentation:** 3 good
**Contribution:** 3 good
**Rating:** 6
**Confidence:** 3

**Summary:**

This paper investigates mechanisms in recurrent neural networks for solving tasks that require long memory representation. The motivation for this work is that organisms need to process signals from a variety of timescales, and discovering mechanisms for timescale adaptation may be useful. Prior work has studied timescales in both single neurons and networks, and this paper aims to study specifically how these timescales shape RNN dynamics and performance on long-memory tasks.

The paper introduces the model: an RNN where each neuron has a trainable leak parameter tau - for higher values of tau, memory is a higher component of the input; for lower tau, the incoming input and recurrent signals dominate. The paper considers dynamics in terms of the single-neuron timescales tau and the network timescales (estimated from the decay rate of the neuron activity's autocorrelation).

Experiments are conducted for two tasks: N-parity and N-DMS. Two curricula are used: single-head, with a single response head that does the task for the input N, and multi-head, which adds a head for every subsequent N during training and therefore does the task for all integers up to N. This mitigates catastrophic forgetting seen in the single-head model as higher N come up in the curriculum.

Results are presented for a variety of research questions. First, performance under different curricula: results here show performance without a curriculum (N alone) - these fail for N > 10. Then, single-head vs. multi-head: multi-head does far better. Next, investigation of the mechanisms underlying long timescales, in which the paper argues that while the single-head curriculum leads to long single-neuron timescales for memory purposes (large, heterogenous tau), the multi-head curriculum leads to shorter tau (and higher tau_net and activating recurrent interconnectivity), suggesting that the memory mechanism is more in the recurrent connections. Finally, ablations are investigated, looking at ablations of neurons with especially short or long tau values, perturbations to weights and timescales, and retraining, for all of which the multi-head approach is superior.

The paper finally takes us through a more detailed version of the related work in the intro, and concludes by suggesting that 1) training on sets of related tasks may lead to more performant and robust networks than training on single tasks and 2) long timescales are necessary for long memory but these can be achieved through connectivity rather than just single neurons.

**Strengths:**

#### Quality
- Experiments are highly intuitive
- The paper is framed around comparison - N-parity and N-DMS, results on which both serve the same goal and also allow comparison on difficulty; single- and multi-head; single-neuron and network-mediated timescales. The results space is therefore very accessible to reason about.
- Results are compelling and analysis is convincing - section 4 has a lot of useful material, and each subsection makes interesting claims. Overall, the results mostly back up the claims.

#### Clarity
- Figure 7 is useful and well-annotated, and a good way of presenting the data
- Sections 2 and 3 is well-written. The model setup and experiments are presented very clearly - I am not confused at any point.
- More involved concepts (tau_net, autocorrelation) are made easy to understand
- Ablation/perturbation/retraining setup is particularly useful to understand the nature of the problem.

#### Originality
Based on related work, this paper does appear to position itself as an experimental realization of theory presented in prior works, leading to a novel and unknown result

#### Significance
The setup is compelling, and the suggested conclusions (1 and 2 from my summary above) are promising and significant if useful.

**Weaknesses:**

#### Quality
- Claims in conclusion seem overstated. This is more a significance issue than a quality issue.
- I would say more experiments are needed, but it's not a matter of specific additional experiments - for what it is, this seems to be a careful and informative set of experiments. It's more that the paper needs considerable expansion, which may also be a significance issue.
- Figure 8: the curves do show a difference but not a drastic one, making it hard to judge how much more robust multi-head is. Significance testing, grounding according to some baselines (including visually on the figure), or comparison to similar results in other contexts may help.

#### Clarity
- While Section 4 has a lot of good info, it is difficult to understand. The writing is prose-based and somewhat meandering and disorganized. The subsection structure is useful, but I think it would help to add subsubsections, especially because the subsections are topic-based rather than claim-based. It would help to add claim-based subsections (or paragraphs) - e.g. in section 4.2, a subsubsection summarizing in a sentence how tau and tau_net change as N increases.
- Aside from fig 7, figures are under-annotated - it takes a lot of effort to understand the message. This could be easily rectified with more visual information. Figure 4a in particular needs better explanation - the x-axes are confusing.
- The term "time-lag" is introduced without clear definition, and it takes a while to get the relevant info.

#### Originality
No weaknesses as far as I am aware.

#### Significance
As stated in the limitations, there is only one testbed in this paper and it is very toy. Unfortunately, the claims of this paper seem to be about improving RNN performance, which means that some kind of scale or breadth is requisite. The motivation connects to biological neural networks, but that stops at the intro - the paper never connects the investigated mechanisms back to biological intelligence in any way after the prose in the intro, let alone through experimentation. It's therefore unclear how significant these results are, though they are intriguing.

**Questions:**

- What other experimental settings/testbeds might be easily accessible for your setup?
- How might this connect back to biological intelligence? I realize this isn't necessarily the focus of the paper, but it is discussed extensively in the intro.
- Section 4.2 goes into this but it's hard to get key conclusions: it's clear that with the multi-head curriculum, single-neuron timescales are not the dominant memory mechanism. What intuition should we take away about the memory mechanism the multi-head curriculum *does* induce? We see that high, less inhibitory network connections are involved - anything more specific?

---

> ### Author Response · Authors · 2023-11-20
>
> We thank the referee for constructive feedback and helpful comments.
> In the following, we go over the questions/ noted weaknesses and describe changes and new analyses.
>
> - Overstatement of the results:
>
> We edited the paper and made our statements more grounded and concrete (e.g., Relevance to Neuroscience, Related Work sections) and extended the limitations section
>
> - Additional experiments:
>
> While we could not generate an entirely new set of experiments in a short time, we propose the general set of requirements these experiments need to satisfy (in the limitation section). Moreover, we described an additional task that would fulfill these requirements (Appendix D).
>
> - Significance of single-head vs multi-head perturbation and retraining (former Fig.8):
>
> We run the significance test for different perturbation strengths and retraining N (Appendix M). For all perturbations, when the perturbation size is very small, there is almost no difference; for larger perturbations, there is a significant difference, and then performance for both networks degrades to the level where they are indistinguishable again.
>
> - Clarity:
>
> We edited section 4 to have paragraph headings and clear claim-based summary sentences, modified figure captions and introduced time lag earlier.
>
> - Different experimental settings/ testbeds:
>
> We added to the limitation section a requirement for the task. Moreover, we added Appendix D, discussing a possible task for future research: pattern generation, which is a common task. We adapt this continuous-time task and turn it into a curriculum for generating the sum of sine waves.
>
> - Relevance for neuroscience:
>
> We see our work as strongly motivated and interpretable in the neuroscience context. We added a new section 5 to better discuss this relevance. In brief, our results can be extended to continuous-time settings relevant to modeling brain networks. Moreover, we discuss that using recurrent connectivity to learn the tasks and develop long timescales aligns well with previous experimental findings.

---

### Official Review · Reviewer_vRYi · 2023-11-10

**Soundness:** 3 good
**Presentation:** 4 excellent
**Contribution:** 2 fair
**Rating:** 5
**Confidence:** 4

**Summary:**

The paper trains RNNs on two tasks that require memory with two curriculum learning types of approaches, one with sequential learning of a single-head RNN and one with multi-head RNN. The former learns a single task sequentially and the latter simultaneously learns multiple closely related tasks that build upon one another simultaneously. The paper shows that these two approaches exhibit different properties in single-neuron and network-mediated timescales.

**Strengths:**

**Originality**
- The paper proposes and explicitly evaluates experiments on single-neuron and network-mediated neuron timescales properties for two overarching tasks, with different value $N$ subtasks, and two curriculum learning type approaches.

**Quality**
- The paper does a good job for the most part on evaluating and showing results on means and standard deviations and some sort of error or measure of spread of each of these for various properties of interest (e.g., $\tau_{\text{net}}$) in Figure 5 as $N$ changes.
- Most claims are supported by empirical results.
- The paper identifies or is inspired by other sources to make certain principled decisions, such as identifying that autocorrelation bias is an issue, and simulating network activity for long periods of time to mitigate this issue. (Mentioned in Page 3)

**Clarity**
- The paper generally does a great job explaining the motivation, related work, preliminaries, some experimental design, and some analysis of results. For example, page 3 contains an excellent explanation of the limitation posed by having non-linear dynamics and how the proposed approach overcomes these limitations through an approximation method that the paper explains as well.
- Figures are generally well made. Some take a little effort to decipher, but I was able to understand all of them. However, accessibility to mildly or significantly colorblind people could be improved by differentiating the single and multi-head N-DMS line plots in Figure 4 and any other similar cases.

**Significance**
- With concrete results on specific tasks, I hope this work inspires people to further explore this area, especially in ways that show that it generalizes as well as cases that deviate from expected behavior with analyses and findings as to potential causes.

---

Note: I've raised my ratings of Soundness from a 2 to a 3, Presentation from 3 to a 4, and of the overall paper from a 3 to a 5 due to the extensive improvements made and clarifications provided by the Authors during the Author Rebuttal phase.

**Weaknesses:**

For Figure S12, it would be helpful to readers to point out that the y-axis is on a log scale, so even though visually the linear fit doesn't look good towards larger $t$, the residual errors are much smaller here because this regime has much smaller values than those for earlier $t$.

In Figure 6, I can infer, but I don't believe it is explicitly stated why N=30 is used as the maximum value for linear regression line fitting. Furthermore, I don't understand why the paper uses N=30 for N-DMS if the reason to the previous question is that N=30 was about the limit for single-head networks on N-parity.

For Figure 8c, I would've liked to see a mean line and N extend to a higher number to see at what relative accuracy each converge and how these compare to one another other.

The Limitations section is lacking significantly, especially with work as general as individual and network neuronal timescales in DNNs. Future work is also lacking, both as a result of lacking limitations as well as no suggestions for future work outside of the limitations section. This section seems like an afterthought that was thrown in to satisfy having "Limitations" of the work mentioned.

With the large amount of related work that is close to the proposed work, it seems like the proposed work is primarily an empirical investigation into very simple tasks and limited approaches. Even equations are generally adapted from another source, and the paper merely needs to plug in its variable symbols to create the equations shown. I think there is novel concrete empirical insights here, but I'm not confident as to how they generalize based on the limited number of experiments provided. I have more confidence in how they generalize based on the related literature's work that forms the basis for which the paper proposes and evaluates the experiments it does. Though the paper is generally well written and presented, I find that it being primarily, or even about entirely, empirical that it is too limited in its experiment design quality, empirical results, and scope to recommend it for acceptance in its current state.

**Questions:**

1. Page 3 states "In the limit case τ → ∞, the neuron’s activity is constant, and the input has no effect." Does this assume convergence as $\tau$ approaches infinity? Are there cases in which this wouldn't hold?

2. The paper uses AIC to select the best fitting model.
  a. Why do you use this information criterion?
  b. Why use this criterion instead of another or others collectively?
  c. For support in using this criteria, consider referencing [1] that used AIC alongside other information criterion to select the best fitting model for time-series data and won the IJCAI'23 Best Paper Runner Up award.

3. Figure 4 caption says "whereas in the multi-head curriculum, having τs fixed at k value is as good as training them." Should this include a specific numerical value or range of values for $k$?

4. On page 5, how is optimal strategy defined in the statement "Furthermore, the multi-head curriculum emerges when training on a multi-objective task without introducing an explicit curriculum (Appendix D), supporting the optimal strategy of this curriculum."

5. In Figure 5c, 5d, do you have insight why the mean and standard deviation for the multi-head networks go up and then back down? I.e., they peak at ~30 and then go down.

6. On page 7, could you further explain the statement "The strong negative weights in single-head networks are required to create stable dynamics in the presence of long single-neuron timescales."

7. For the "retraining" (Perhaps a slight typo) experiment details on Page 8 and 9, why use 20 epochs?

[1] Pasula, P. (2023). Real World Time Series Benchmark Datasets with Distribution Shifts: Global Crude Oil Price and Volatility. arXiv preprint arXiv:2308.10846.

**Details Of Ethics Concerns:**

I have no ethics concerns for this work.

---

> ### Author Response · Authors · 2023-11-20
>
> We thank the referee for motivating questions and suggestions. In the following, we address noted weaknesses and suggestions.
>
> - Editing for colorblind:
>
> Thank you for the suggestion. We changed the figures such that in the panels where the single- and multi-head lines for different experiments are present together, the multi-head lines are dashed (Fig. 3b, Fig. 4b). In all other cases, we checked in Black and White that the lines are distinguishable.
>
> - Figure S12:
>
> Thank you, we implemented your suggestion (new Fig. S2).
>
> - Figure 6:
>
> Indeed, we fitted the linear relationship only for small N (N =20). To make it easier to identify, we added the lines as a guide for the eye. The reason for this is to show that the scaling of the dimensionality in the N-DMS task is not linear: for the points with N>>20, the line does not fit, and growth is slower. In the updated version of Fig.6 and new Fig.S20, we present the fits for different ranges.
>
> - Figure 8c (now 7e):
>
> We now include the chance-level accuracy in the panel. We retrained the networks for higher N and observed for both single and multi-head curricula, a convergence towards the chance level. Moreover, we calculated the p-values and presented them in Appendix K to demonstrate that there is a significant difference between single- and multi-head networks.
>
> - Relation to the previous work and limitations:
>
> We substantially edited the related work and limitations sections to highlight the significance of our findings in comparison to previous works and its potential extensions for future studies. In brief, although there were theoretical studies analytically deriving timescales for simple networks, those studies were limited to the specific (and simplistic) connectivity and investigated only the relationship between the topology features and local timescales, often without discussing their functional relevance. Here, we extend those findings and relate the dynamical properties directly to learning and computations relevant to working memory tasks, bridging theoretical mechanistic concepts from computational neuroscience to functional concepts in ML. We also show that the multi-head curriculum improves not only the training of RNNs but also of GRUs and  LSTMs.
>
> Questions:
>
> - Convergence:
>
> The equation is  $r(t) = \[(1 - 1/τ) \cdot r(t-1) + \sigma(\text{Input}) / \tau \]_a$
>
> If $\tau$ → ∞ , then $1/\tau$ →  0, so the equation becomes:
>
>  $r(t) = [r(t-1)]_a$
>
> If $r(0) \geq 0$, then r(t) = r(0) for all t (since $[x]_a = x$, for $x\geq 0$ and $[x]_a = a \cdot x$, for x < 0 )
>
> If $r(0) < 0$, then $r(t) = a^t  \cdot r(0)$ →0, since $a << 1$
>
> Thus as $\tau$ → ∞, we have convergence
>
> - Model selection:
>
> Thank you for the suggestion of the relevant literature for the choice of model selection criterion. We referred to it and also tested BIC and found that there is above 95% agreement between AIC and BIC (Fig. S1).
>
> - Figure 4 (meaning of k):
>
> Thank you for noting, we edited the caption and added a more detailed explanation about k interpretation in new section 5 and Appendix C.
>
> - Training multi-N without curriculum statement:
>
> We rewrote it as:  “Furthermore, training directly on a multi-N task without using an explicit curriculum results in the emergence of the multi-head curriculum: networks learn first to solve small-N tasks and then large-N tasks (Appendix I), supporting the use of the multi-head step-wise strategy.”
>
> - Figure 5 - Behavior of timescales for large N:
>
> For large N, the network may need strategies that go beyond increasing the timescales of individual neurons (measured by tau and tau_net).  These solutions possibly employ coordination across multiple neurons to generate slow dynamics on the population level rather than the activity of individual neurons.  As a proxy for such coordinations, we measure the timescale of population dynamics (summed activity of all neurons) tau_pop and show that for all tasks and curricula, tau_pop grows with N (Fig. S4).
>
> - Meaning of the strong inhibitory weights:
>
> In the single-head curriculum, the long taus make the influence of recurrent and external inputs larger (equ. 1). Thus, a perturbation can lead to a very large response, if inhibition does not stop it. We illustrated this stability issue by perturbing the inhibitory weights (Appendix  J., Fig. S19). For single-head networks, decreasing the inhibition leads to the rapid explosion of network activation, whereas for multi-head networks changes are minimal.
>
> - Retraining:
>
> Indeed, we took 20 epochs for retraining. The reason is to capture the speed of convergence, which would indicate the benefit the training has from the initialization with a network trained for N=16 (single-head), or N=2,..., 16 (multi-head). Generally, many networks can train to solve N=17 from scratch, so both initializations would converge to the perfect performance if given enough time. We tested retraining for 50 epochs, which produced qualitatively similar results.

---

> > ### Comment · Reviewer_vRYi · 2023-11-23
> >
> > Thank you, Authors, for your detailed feedback.
> >
> > **Editing for colorblind:** Thank you for adjusting these figures so that they are more accessible to those who may have a form of colorblindness or similar.
> >
> > **Figure S12:** Looks good. Your paper will be clearer to some readers now. For example, I had to reason about why there is such a strong linear fit even though visually it isn't evident without realizing and connecting that the y-axis is log scaled.
> >
> > **Figure 6:** Thanks for adding the additional fits. It's more informative and transparent to readers.
> >
> > **Figure 8c (now 7e):** Thanks, this is interesting.
> >
> > **Relation to the previous work and limitations:** Great to see that you've added significantly to the Limitations section with limitations, future work suggestions, a potential experiment in Appendix D and references for future work suggestions and another potential experiment.
> >
> > **Convergence:** Thanks for showing this rigorously.
> >
> > **Model selection:** Thanks for looking at BIC as well and including results showing that there is over 95% overlap between which model is chosen as well as the note about the HQC. AIC is indeed the often the most used, but it's nice to see that a different model selection criterion that is also often used has strong overlap with AIC.
> >
> > **Figure 4 (meaning of k):** Looks good.
> >
> > **Training multi-N without curriculum statement:** Looks good.
> >
> > **Figure 5 - Behavior of timescales for large N:** Thanks for the explanation and the empirical results. Note: for me, the paper's clickable \refs For Fig. S4 are taking it to Figure 4 instead of Figure S4. This is something you may want to verify on your end.
> >
> > **Meaning of the strong inhibitory weights:** Interesting! Thanks for including this.
> >
> > **Retraining:** Thanks for looking into this. I do also see the new Figure S13 in the Supplementary material that seems to show results on this.
> >
> > ---
> >
> > Thank you to the authors for a very diligent follow-up in terms of clarifying ambiguities, taking Reviewer review points into consideration and clarifying ambiguities, significantly modifying the paper, and running additional experiments to address weaknesses or answer questions raised by Reviewers. I will be raising my score from a 3 to a 5.
> >
> > I will need more time to consider raising the score to a 6, which is mostly limited by uncertainty as to applicability to tasks with even the remote complexity of almost all real-world ones. However, the huge improvements in Limitations, Related Works, Future Work and the thorough and strong experimental analyses are large enough for me to happily raise my rating by 2 points. Nice work!

---

### Author Response · Authors · 2023-11-20

We thank the referees for their constructive comments and helpful suggestions. We have performed new analyses, edited the manuscript accordingly (the revised version is uploaded) to address their concerns, and provided detailed responses to each referee.

Summary of the most important changes:

- Provided a more clear description of the significance of our findings compared to previous work, extended the limitations section, and proposed a new experiment as a future direction.

- Highlighted the implications of our findings for the neuroscience community by adding a new section dedicated to it (Section 5)  and including new experiments.

- Provided a clearer description of $\tau_\mathrm{net}$ and included new analyses of population activity timescales ($\tau_\mathrm{pop}$) that strengthen our claims about the role of network-mediated mechanisms in shaping timescales (Fig. S4).

- Added new analyses on the role of strong inhibition in the stability of single-head networks (Appendix  J).

- Added significance tests’ results for comparing single- and multi-head networks’ statistics.

---

### Meta-Review · Area_Chair_2RyN · 2023-12-04

**Metareview:**

This paper examines what happens in recurrent neural networks (RNNs) when the time-constants of individual neurons are also optimized along with other network parameters. The RNNs are trained on N-parity and N-delayed match-to sample tasks. The authors find that whether the networks learn to use extended time-contstants or recurrent connections to store information over time depends on the nature of the training (whether a single N is used or multiple Ns). They further show that the show that the multi-N curriculum improves training speed, stability to perturbations, and generalization.

The reviewers were overall positive, but raised several concerns related to clarity, articulation of limitations, relatively toy tasks, and some other items. However, post-rebuttal, the reviewers were sufficiently convinced to bring their scores up, and the final tally was 5,6,8,8, which places this paper clearly in accept territory. The AC felt that the paper clearly provides an interesting demonstration of a principle that could be of use to the computational neuroscience community.

**Justification For Why Not Higher Score:**

This paper is interesting, but it is ultimately relatively toy and mostly of interest to computational neuroscientists. Thus, a spotlight at ICLR is probably not appropriate.

**Justification For Why Not Lower Score:**

The reviewers all agreed that there is value in this paper, and I agreed with them. It is well written and up to a standard for acceptance.

---

### Decision · Program_Chairs · 2024-01-16

Accept (poster)